# EventMG: Efficient Multilevel Mamba-Graph Learning for Spatiotemporal Event Representation

Sheng Wu[1]    Lin Jin[1]    Hui Feng[1,2] *    Bo Hu[1,2]

[1] College of Future Information Technology, Fudan University

[2] State Key Laboratory of Integrated Chips and Systems, Fudan University

## Abstract

Event cameras offer unique advantages in scenarios involving high speed, low light, and high dynamic range, yet their asynchronous and sparse nature poses significant challenges to efficient spatiotemporal representation learning. Specifically, despite notable progress in the field, effectively modeling the full spatiotemporal context, selectively attending to salient dynamic regions, and robustly adapting to the variable density and dynamic nature of event data remain key challenges. Motivated by these challenges, this paper proposes EventMG, a lightweight, efficient, multilevel Mamba-Graph architecture designed for learning high-quality spatiotemporal event representations. EventMG employs a multilevel approach, jointly modeling information at the micro (single event) and macro (event cluster) levels to comprehensively capture the multi-scale characteristics of event data. At the micro-level, it focuses on spatiotemporal details, employing State Space Model (SSM) based Mamba, to precisely capture long-range dependencies among numerous event nodes. Concurrently, at the macro-level, Component Graphs are introduced to efficiently encode the local semantics and global topology of dense event regions. Furthermore, to better accommodate the dynamic and sparse characteristics of data, we propose the Spatiotemporal-aware Event Scanning Technology (SEST), integrating the Adaptive Perturbation Network (APN) and Multidirectional Scanning Module (MSM), which substantially enhances the model's ability to perceive and focus on key spatiotemporal patterns. By employing this novel collaborative paradigm, EventMG demonstrates the ability to effectively capture multi-level spatiotemporal characteristics of event data while maintaining a low parameter count and linear computational complexity, suggesting a promising direction for event representation learning.

## 1 Introduction

Event cameras, a novel class of vision sensors that asynchronously record pixel brightness changes to generate high-speed, sparse data streams [1], have garnered significant attention in recent years due to their unique capabilities. Unlike traditional cameras with fixed time exposure, event cameras offer microsecond-level temporal resolution and a dynamic range exceeding 120 dB—far surpassing the 60-70 dB of conventional cameras [2, 3]. These attributes provide substantial advantages in challenging scenarios such as high-speed motion, low-light conditions, and high-contrast environments [4].

Despite their distinct advantages, event cameras pose significant challenges for data processing. The asynchronous generation of events in response to changes in pixel brightness results in outputs with irregular spatiotemporal properties, high temporal resolution, and large instantaneous data volumes

---

*Corresponding author.

39th Conference on Neural Information Processing Systems (NeurIPS 2025).

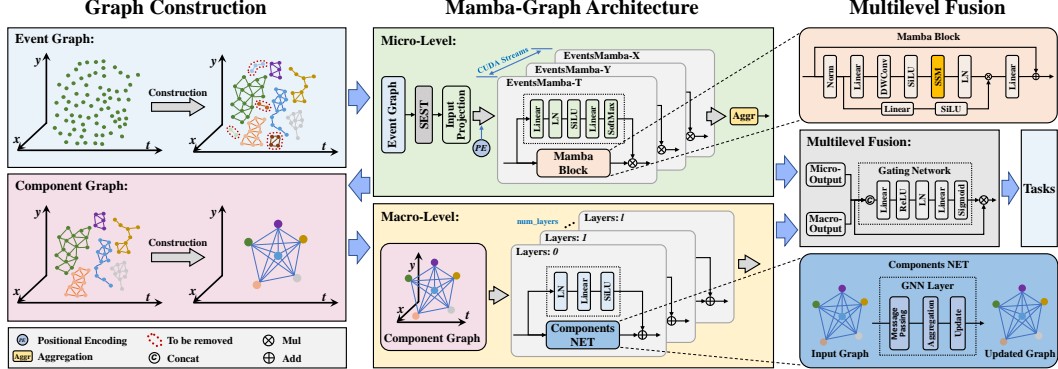

Figure 1: **The overall workflow of EventMG** comprises three main modules: 1) **Graph Construction**, which constructs the micro-level Event Graph and macro-level Component Graph; 2) **Mamba-Graph Architecture**, which performs spatiotemporal modeling at both levels; and 3) **Multilevel Fusion**, which integrates outputs across levels to produce the final spatiotemporal representation for downstream tasks.

[5]. These characteristics make it difficult to directly apply traditional algorithms, which are typically designed for regular and dense data, to event-based modeling and representation learning.

A straightforward strategy to address the above challenges is to convert the event stream into frame-based dense representations (e.g., event counting or time surfaces [6, 7]). This conversion facilitates the application of established frame-based processing methods, including Transformer-based architectures, which have shown improved performance on certain tasks [8, 9]. However, this event-to-frame conversion undermines the core microsecond-level spatiotemporal precision of event data, introducing potential motion blur in high-speed scenarios. It also increases data density and redundancy in non-event regions, negating the intrinsic sparsity benefits of event data [10], thereby limiting the full exploitation of its spatiotemporal richness.

To retain the spatiotemporal precision and sparsity, recent methods directly process sparse event streams using point-based representations. Techniques based on point clouds [11–14], Graph Neural Networks (GNNs) [15–18], and Spiking Neural Networks (SNNs) [19–22] have emerged to avoid dense conversions. To further capture the dynamic context and spatiotemporal dependencies of event streams, Transformer-based architectures have been integrated into sparse processing pipelines [23, 24]. Although self-attention enables effective modeling of spatiotemporal features of dynamic event streams, its quadratic complexity ($O(N^2)$) poses scalability challenges for long sequences, resulting in high computational costs and large models.

This paper proposes EventMG, a lightweight and efficient multilevel spatiotemporal representation learning architecture specifically tailored for event data, to address the limitations of existing methods, particularly concerning the spatiotemporal precision, representation adequacy, and computational cost. EventMG achieves comprehensive modeling by synergistically capturing both micro-level (individual event spatiotemporal details) and macro-level (event cluster structure and semantics) information, enabling systematic feature extraction and fusion while preserving spatiotemporal precision. At its core, EventMG features an innovative Mamba-Graph structure: Mamba [25], leveraging State Space Models (SSMs) [26–28], efficiently models long-range spatiotemporal dependencies within large-scale events at linear ($O(N)$) computational cost, capturing fine-grained dynamics. Concurrently, Component Graphs [29] are employed to effectively encode local semantics and global topological structures within dense event regions, thus realizing a deep integration of event points and their relational graph context. Furthermore, to specifically adapt Mamba for the unique sparsity and dynamism inherent in event streams, we design the Spatiotemporal-aware Event Scanning Technology (SEST). Integrating an Adaptive Perturbation Network (APN) and a Multi-directional Scanning Module (MSM), SEST innovatively enhances the Mamba framework, significantly improving its capacity to perceive and model complex spatiotemporal patterns native to event data.

Experimental evaluation on representative dynamic tasks indicates that EventMG delivers performance comparable to state-of-the-art heavyweight models, but at a lower computational and parametric cost. The results underscore the potential of our proposed architectural paradigm, which seeks to strike a deliberate balance between computational efficiency, representational power, and respect for the intrinsic properties of event data.

The main contributions of this study are summarized as follows:

- We introduce a novel Mamba-Graph collaborative architecture that combines the strengths of SSM-based and graph-based approaches, efficiently modeling dynamic event data from micro-level spatiotemporal details to macro-level cluster patterns.
- We develop the Spatiotemporal-aware Event Scanning Technology (SEST) that specifically adapts SSM-based Mamba to the unique sparse and irregular spatiotemporal characteristics of event streams, significantly enhancing their capacity to model complex dynamic patterns.
- We present EventMG, a lightweight and efficient spatiotemporal event representation architecture achieving an effective balance between high performance and minimal computational overhead (linear complexity) without requiring pre-training, across diverse dynamic tasks.

## 2 Related Work

The unique asynchronous sparse data from event cameras poses challenges for efficient representation learning and modeling, driving diverse processing strategies with varying efficiency and performance. This chapter reviews key event data representation approaches and spatiotemporal sequence modeling architectures, analyzing their characteristics and limitations.

### 2.1 Event Data Representation Strategies

**Frame-based Representations:** A common strategy to exploit mature frame-based methods (e.g. CNNs, Vision Transformers) is to convert asynchronous, sparse event streams into dense, image-like representations by accumulating events over fixed counts [30] or time windows [31], effectively projecting and compressing temporal information into 2D or 3D tensors. This enables direct use of existing architectures but compromises microsecond-level precision and destroys sparsity, leading to motion blur, redundant computation, and low efficiency [10, 32, 33]. Furthermore, processing these dense representations often requires heavyweight models, increasing latency [23], thus fundamentally limiting the exploitation of event camera advantages.

**Sparse Representations:** Directly processing sparse events preserves spatiotemporal fidelity, better aligning with event data characteristics. Point cloud methods [11, 12] treat events as spacetime points, processed by adapted point networks or specialized kernels [13, 14], but must contend with unordered sets and neighborhood definitions. Graph-based methods [15–17] use GNNs on constructed event graphs to capture structure, offering flexibility but facing challenges in effective graph construction, especially under event noise [34] and high dynamics. Spiking Neural Networks (SNNs) [35–37], bio-inspired asynchronous models well-suited to event data due to their spike-driven mechanisms [4, 38], offer high energy efficiency potential [21, 39] but present training challenges, representing a distinct research path. Although these sparse methods preserve data fidelity, effectively modeling long-range spatiotemporal dependencies in large-scale data remains the key challenge to deeply understanding dynamic scenes and learning insightful spatiotemporal representations.

### 2.2 Spatiotemporal Sequence Modeling

**Transformer-based Methods:** Transformers [40–43] have been introduced into event processing due to its strong representation capabilities and ability to model global dependencies through the self-attention mechanism. These methods have been applied to frame-based sequences [8, 9, 44] and directly to sparse event, point cloud, or graph node feature sequences [33, 45, 23, 24, 46]. However, the quadratic complexity $O(N^2)$ of self-attention limits scalability, making it challenging to handle large-scale, high-resolution event data in resource-constrained scenarios.

**SSM-based Methods:** To address Transformer inefficiency in long-sequence modeling, State Space Models (SSMs) have emerged as a promising alternative [26–28, 47, 48]. Mamba [25] enables efficient long-range modeling with linear complexity and hardware-friendly properties, rivaling Transformers. However, applying Mamba to event streams is challenging due to sparsity, asynchrony, and dynamics. EventMG addresses this by introducing APN and MSM modules to improve robustness and spatiotemporal modeling while fully leveraging Mamba's $O(N)$ efficiency.

In summary, EventMG is designed to effectively address the challenges of processing event streams. Building upon the Mamba enhancements (APN and MSM), EventMG further innovates by estab-

lishing a Mamba-Graph collaborative architecture. This unique combination efficiently implements multilevel modeling, capturing information from micro to macro scales and local to global contexts. As a pre-training-free, end-to-end framework, EventMG strikes an excellent balance between performance and lightweight design.

## 3  Methodology

This section details our proposed EventMG, a lightweight end-to-end architecture for efficiently learning multi-level spatiotemporal representations from event streams. We begin in Section 3.1 with a high-level overview of the model's overall architecture. The subsequent subsections then delve into its core modules: micro-level representation learning (Section 3.2), macro-level representation learning (Section 3.3), and the final multi-level fusion strategy (Section 3.4).

### 3.1  Overall Architecture

The overall architecture of EventMG is a multi-level information processing pipeline designed to simulate the "event→object→scene" hierarchy inherent in event data. This pipeline deeply fuses local event details with global context through three core stages: micro-level modeling, macro-level aggregation, and multi-level feedback and fusion. The entire workflow is illustrated in Figure 1.

Specifically, the process begins at the micro-level, where input event slices are first constructed into a sparse graph and then processed by the Spatiotemporal-aware Event Scanning Technology (SEST) and the EventsMamba module to extract fine-grained node features encapsulating long-range dependencies. Subsequently, at the macro-level, the model leverages these micro-level features to construct a Component Graph and perform information aggregation, distilling the macro-level feature that represents the global scene. Finally, through the serial and mutually-feeding mechanism, the macro-level feature is fed back to enhance each micro-level representation, generating a final output that is both detailed and context-aware. This design fully embodies our core philosophy of "starting from the micro, abstracting to the macro, and then guiding the micro with the macro."

### 3.2  Micro-level: Representation based on Events

The processing in EventMG begins at the micro-level, focusing on individual events and their local spatiotemporal context. The core objective at this level is to model fine-grained dynamics and high temporal resolution inherent in event data.

#### 3.2.1  Event Graph Construction

At the micro-level, we first process the continuous event stream. Following established methods [10, 4], we employ a sliding window strategy with a fixed number of events, N, to segment the stream into multiple event slices. This approach ensures that each slice contains a stable amount of information and can adapt to the scene's activity intensity, providing standardized inputs for subsequent processing.

Within each event slice, every individual event is represented as a event node $v_i = (x_i, y_i, t_i, p_i)$, which encapsulates its spatial coordinates, timestamp, and polarity information [4]. To capture local relationships between events, a basic event graph $G = (V, E)$ is constructed, where $V$ is the set of event nodes. We employ an efficient sparse graph construction strategy that establishes connections by finding at most $k$ neighbors for each node, rather than constructing a dense adjacency matrix. This method ensures the graph's storage and computational complexity is $O(Nk)$, maintaining a linear relationship with the number of events, $N$. Consequently, the memory footprint is predictable and strictly controllable, which provides a fundamental guarantee for processing large-scale event streams. The edges $E$ of the graph are established based on spatiotemporal proximity, as defined below:

$$E(v_i, v_j) = \begin{cases} 1 & \text{if } \sqrt{\|s_i - s_j\|_2^2 + c^2(t_i - t_j)^2} \leq \delta_{st}, \\ 0 & \text{otherwise}, \end{cases} \tag{1}$$

where an edge exists if the spatiotemporal distance, calculated as the L2 norm of the spatial component $\|s_i - s_j\|_2$ and the scaled temporal component $c|t_i - t_j|$, does not exceed the threshold $\delta_{st}$.

Although graph construction defines the event's topological structure, the initial node features contain limited information and are insufficient for describing complex local dynamics. To address this, we then introduce a lightweight Event Graph Neural Network (GNN) for local feature enhancement. The module updates and enriches the representation of each node $v_i$ by aggregating information from its local spatiotemporal neighborhood $\mathcal{N}(v_i) = \{v_j \mid E(v_i, v_j) = 1\}$, thereby transforming the original, discrete event point cloud into a graph rich with local features. This preliminary feature distillation process provides high-quality input node features for subsequent operations.

### 3.2.2 Spatiotemporal-aware Event Scanning Technology (SEST)

To efficiently model long-range spatiotemporal dependencies at the micro level, we adopt the SSM-based Mamba model [25]. However, its standard scanning struggles with the sparsity, asynchrony, and dynamics of event data. To address this, we propose the Spatiotemporal-aware Event Scanning Technology (SEST), which integrates APN and MSM modules to optimize event arrangement and scanning (see Figure 2), enabling Mamba to better adapt to the uniqueness of event streams.

**Adaptive Perturbation Network (APN):** Although graph construction defines the event's topological structure, the initial node features contain limited information. To address this, we first employ a lightweight Event Graph Neural Network (GNN) for local feature enhancement. The GNN module updates and enriches the representation of each node $v_i$ by aggregating information from its local spatiotemporal neighborhood $\mathcal{N}(v_i) = \{v_j \mid E(v_i, v_j) = 1\}$, thereby transforming the original, discrete event point cloud into a graph rich with local features. This preliminary feature distillation process provides high-quality input node features, denoted as $\mathbf{h}$, for subsequent operations.

Subsequently, the APN module enhances the model's adaptability to event data and further optimizes the event arrangement through a learnable coordinate perturbation mechanism. It consists of two complementary paths, *Local* and *Global*, to jointly optimize the features:

$$\mathbf{h}_v^{\text{local}} = \text{Conv1D}(\mathbf{h}), \tag{2a}$$

$$\mathbf{h}_v^{\text{global}} = \text{MLP}_{\text{global}}(\mathbf{h}), \tag{2b}$$

$$(\Delta x, \Delta y, \Delta t) = \text{Tanh}\big(\text{MLP}_{\text{final}}(\mathbf{h}_v^{\text{local}} + \mathbf{h}_v^{\text{global}})\big), \tag{2c}$$

where, the *Local* path is processed by a convolution (Conv1d), which operates on the sequence of node features sorted by timestamp $t$. Through its sliding convolutional kernel, it efficiently captures short-term temporal dynamics between an event and its temporally adjacent neighbors. Complementing this, the processing in the *Global* path targets all nodes, aiming to extract global feature patterns independent of the sequential context. By fusing the outputs of these two paths, the APN module computes the spatiotemporal perturbations $(\Delta x, \Delta y, \Delta t)$ based on a comprehensive judgment of both local temporal dynamics and global feature patterns. These perturbation values are then applied to the original coordinates to generate the perturbed positions $(x_i', y_i', t_i')$:

$$x_i' = x_i + \Delta x, \quad y_i' = y_i + \Delta y, \quad t_i' = t_i + \Delta t. \tag{3}$$

The core objective of this adaptive coordinate adjustment is not merely to "correct" the spatiotemporal positions of events, but rather to learn a more discriminative Effective Coordinate Space for the subsequent MSM scanning module. This adaptive coordinate adjustment serves two main purposes: (1) It enhances the expression of event dynamics by refining local spatiotemporal node arrangements (e.g., promoting effective aggregation or dispersion in dense regions), thereby improving the model's robustness to noise and variation. (2) It ensures sufficient distinction across the $x$, $y$, and $t$ dimensions, laying a discriminative foundation for the subsequent multidirectional scanning.

**Multidirectional Scanning Module (MSM):** To overcome the limitations of unidirectional scanning (e.g., the time-based sorting in standard Mamba) in capturing the irregular, multidimensional structure of event data, we introduce the Multidirectional Scanning Module (MSM). The primary function of this module is to construct a set of multidirectional event sequences in preparation for subsequent long-range dependency modeling. It utilizes the effective coordinates optimized by APN to sort the node set $V_i$ along multiple fundamental dimensions $d_k$, thereby generating a set of sequences $\mathcal{S}$. These scanning directions typically include forward and backward scans along the $t$, $x$, and $y$ axes, i.e., the direction set $D = \{t^+, t^-, x^+, x^-, y^+, y^-\}$.

$$\mathcal{S} = \{S_k = \text{Scan}(V_i, d_k) \mid d_k \in D\}. \tag{4}$$

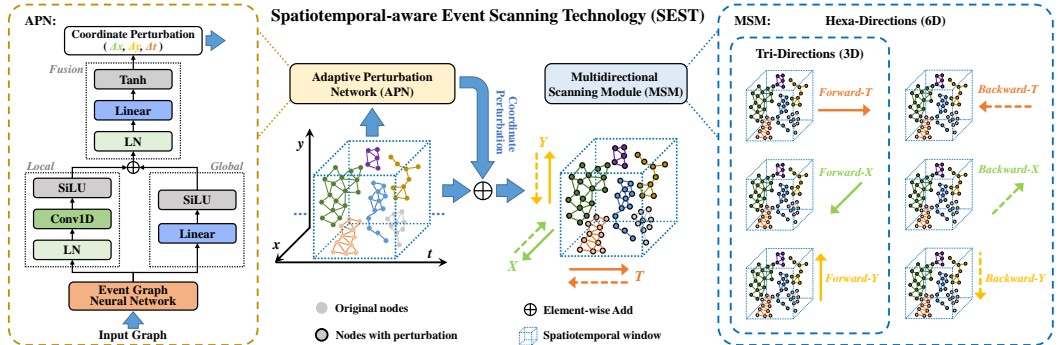

Figure 2: **The Spatiotemporal-aware Event Scanning Technology (SEST)** includes two modules: Adaptive Perturbation Network (APN) and Multidirectional Scanning Module (MSM).

In summary, APN and MSM work in synergy to implement a data-driven, adaptive scanning strategy: APN first learns a task-driven effective coordinate space that is born for scanning, after which MSM performs multidirectional scans within this optimized space. The final output of this process is an optimized set of events $\mathcal{S}$, containing multiple sequential perspectives, which serves as the input for the core modeling module in the next stage.

### 3.2.3 Micro-level Spatiotemporal Modeling

We introduce EventsMamba as our core micro-level modeling module, leveraging Mamba [25], an efficient state-space model, to process large-scale event streams. This module takes as input the set of multidirectional sequences $\mathcal{S}$ generated by MSM from APN-optimized coordinates.

By applying the Mamba core processor along these diverse sequential directions, EventsMamba captures long-range spatiotemporal dependencies and dynamic patterns from multiple perspectives. Finally, the processed results from all sequences are fused into the final fine-grained features $\mathbf{h}_v$ via an aggregation function, as follows:

$$\mathbf{h}_v = \underset{S_k \in \mathcal{S}}{\text{Aggr}} \left( \text{Mamba}(S_k) \right) \tag{5}$$

Combined with Mamba's inherent $O(N)$ linear complexity, this design efficiently extracts node features $\mathbf{h}_v$ that capture detailed event-level dynamics and interactions.

## 3.3 Macro-level: Representation based on Components

Although micro-level modeling captures fine-grained spatiotemporal dynamics, it often lacks awareness of global structure and large-scale patterns, resulting in incomplete representations. To address this, we introduce a macro-level representation strategy based on Component Graphs to effectively extract and organize such macro-structural information.

### 3.3.1 Component Graph Construction

Macro-level modeling begins by identifying and abstracting key structural units in the event stream. Event-dense regions—such as object edges, texture boundaries, or areas with strong luminance contrast—not only concentrate events in space and time but also carry important macro-structural and semantic information. Motivated by their structure, these regions—often forming strongly connected subgraphs—are formalized via the graph-theoretic notion of connected components [49]:

$$
\begin{aligned}
&V_C \subseteq V, \quad E_C \subseteq E, \\
&\forall v_i, v_j \in V_C, \text{ there exists a path } P_{ij}, \\
&\forall v_k \in (V \setminus V_C), \text{ no path connects } v_k \text{ to } V_C, \\
&|V_C| \geq n_{\min}.
\end{aligned}
\tag{6}
$$

A connected component is defined as a maximal subgraph where any two nodes are path-connected and no external node is connected to it. To exclude insignificant regions that may correspond to noise,

we impose a minimum node count threshold $n_{\min}$, retaining only components with $|V_C| \geq n_{\min}$. Each valid component $C$ corresponds to a subgraph $G_C = (V_C, E_C)$.

Next, we construct the Component Graph as a macro-level structural representation from valid connected components, each abstracted as a supernode. Given the typically small number of supernodes representing high-level relationships, we adopt a fully connected strategy. The feature of each supernode $\mathbf{h}_C$ is obtained by average pooling over its constituent event nodes $\mathbf{h}_v$ within the component, which effectively captures essential characteristics while reducing processing complexity:

$$\mathbf{h}_C = \frac{1}{|V_C|} \sum_{v \in V_C} \mathbf{h}_v. \tag{7}$$

The component graph offers three core benefits: (1) It provides a macro-level, structured view of event stream, simplifying the underlying event graph by modeling scene structure and dynamics at a higher level. (2) By selecting valid components, it denoises the data and highlights salient structures, yielding a more robust basis for downstream feature learning. (3) Together with micro-level representations, it forms the foundation of EventMG's multilevel architecture, enabling cross-level interaction.

### 3.3.2 Macro-level Spatiotemporal Modeling

After constructing the Component Graph $G_C$, we introduce the ComponentsNet module for macro-level spatiotemporal modeling. As the key processing unit at this level, ComponentsNet captures structural interactions between supernodes and scene-level dynamics. Leveraging Graph Neural Networks (GNNs), the module performs message passing and node updates over the component graph, aggregating information from connected supernodes to model their dependencies. This produces high-level representations of the scene's overall state and dynamics, offering essential macro-context for guiding micro-level features in subsequent processing.

### 3.4 Multilevel Fusion for Comprehensive Representation

To obtain a unified and informative representation, EventMG introduces an adaptive multilevel fusion mechanism based on learnable gating. The macro-level feature $\mathbf{h}_C$ is first mapped via a learnable layer $f_{map}$ and assigned to each micro-level event node $v \in V_C$ as macro-context $\mathbf{h}_{C \to v} = f_{map}(\mathbf{h}_C)$. A gating network then adaptively fuses the node's micro feature $\mathbf{h}_v$ with its macro-context $\mathbf{h}_{C \to v}$, producing the final fused feature $\mathbf{h}_v^{\text{fused}}$:

$$g_v = \sigma(\text{MLP}([\mathbf{h}_v, \mathbf{h}_{C \to v}])), \tag{8a}$$

$$\mathbf{h}_v^{\text{fused}} = g_v \odot \mathbf{h}_v + (1 - g_v) \odot \mathbf{h}_{C \to v}, \tag{8b}$$

where $g_v$ is the gating weight learned by a multi-layer perceptron (MLP), $[\cdot, \cdot]$ denotes feature concatenation, $\sigma$ is the Sigmoid function, and $\odot$ is element-wise multiplication.

This multilevel fusion mechanism integrates micro-level details with macro-level structure, enabling fine-grained features to be contextually guided and macro patterns refined by local specifics. This bidirectional interaction results in spatiotemporal representations that are robust, comprehensive, and discriminative, effectively capturing fast local changes as well as global structural shifts.

## 4 Experiments

In this section, we evaluate EventMG on two representative tasks with high spatiotemporal dynamics: **Object Detection** and **Action Recognition**, demonstrating its effectiveness in learning spatiotemporal representations from dynamic event streams.

### 4.1 Datasets and Evaluation Metrics

**Object Detection** is evaluated on two traffic scene event camera datasets using three standard metrics.

- **Gen1 Dataset** [50]: The Gen1 Automotive Detection Dataset comprises over 39 hours of $304 \times 240$ event video, captured across urban, highway, and rural traffic scenarios. It contains over 255,000 manual annotations for pedestrians and cars, annotated at a frequency of 1-4 Hz.

Table 1: **Comparison on Gen1 and 1Mpx Datasets.** Performance is reported in terms of parameters (Params), mean Average Precision (mAP), and backbone inference time per batch (Time). A star (*) indicates estimated values. Best results are in bold; second-best are underlined.

| Method | Backbone | Params (M) | Gen1 | | 1Mpx | |
|---|---|---|---|---|---|---|
| | | | mAP (%) | Time (ms) | mAP (%) | Time (ms) |
| Asynet [32] | Sparse CNN | 11.4 | 14.5 | - | - | - |
| RRC-Events [63] | CNN | >100* | 30.7 | 21.5 | 34.3 | 46.4 |
| YOLOv3 Events [64] | CNN | >60* | 31.2 | 22.3 | 34.6 | 49.4 |
| AEGNN [15] | GNN | 20 | 16.3 | - | - | - |
| Spiking DenseNet [22] | SNN | 8.2 | 18.9 | - | - | - |
| ERGO-12 [44] | Transformer | 59.6 | 50.4 | 69.9 | 40.6 | 100.0 |
| RED [51] | CNN + RNN | 24.1 | 40.0 | 16.7 | 39.3 | 24.1 |
| RVT-B [8] | Transformer + RNN | 18.5 | 47.2 | 10.2 | 47.4 | 11.9 |
| Nested-T [65] | Transformer + RNN | 22.2 | 46.3 | 25.9 | 46.0 | 33.5 |
| GET-T [9] | Transformer + RNN | 21.9 | 47.9 | 16.8 | **48.4** | 18.2 |
| S5-ViT-B [66] | Transformer + SSM | 18.2 | 47.4 | **8.16** | 47.2 | **9.57** |
| **EventMG (ours)** | GNN + SSM | **1.96** | **53.7** | 8.66 | 47.4 | 9.63 |

- **1Mpx Dataset** [51]: The 1 Megapixel Automotive Detection Dataset provides over 14 hours of high-resolution event video and 25 million annotations for cars, pedestrians, and two-wheelers, making it ideal for developing advanced detection models in dynamic traffic scenes.

- **Evaluation Metrics**: We evaluate model performance using three primary metrics: 1) Total Parameters, indicating model complexity; 2) Mean Average Precision (mAP@0.5:0.95), computed with COCO [52] toolkit to asses accuracy; 3) Mean inference Time, measuring runtime efficiency.

**Action Recognition** is assessed on four representative datasets using three evaluation metrics.

- **THU$^{\text{E-ACT}}$-50-CHL Dataset** [53]: The dataset is a very challenging dataset with 50 action classes and 2330 recordings, capturing actions of 18 students from various viewpoints in long corridor and hall scenes using a DAVIS346 camera ($346 \times 260$), with each clip lasting 2–5 seconds.

- **DVS Action Dataset** [54]: The dataset contains 10 actions performed by 15 subjects in an office environment, captured by a DAVIS346 event camera at $346 \times 260$ resolution.

- **HMDB51-DVS & UCF101-DVS** [55]: Event-based versions of HMDB51 [56] and UCF101 [57], converted using a DAVIS240 camera. HMDB51-DVS includes 6,766 clips across 51 classes, while UCF101-DVS comprises 13,320 clips from 101 classes. Both use a resolution of $320 \times 240$.

- **Evaluation Metrics:** We evaluate model performance based on three key metrics: 1) Parameters (Params), reflecting model size and complexity; 2) GFLOPs, measuring the computational cost; and 3) Accuracy (Acc), assessing predictive precision on the recognition task.

Notably, while action recognition datasets such as DVS128 Gesture [58] and Daily DVS [39] are widely used, most advanced methods already achieve near-saturation accuracy close to 100%. To better assess EventMG, we conduct experiments on the above more discriminative datasets.

## 4.2 Implementation Details

EventMG is built upon a hierarchical multi-stage architecture designed to extract comprehensive spatiotemporal features. The implementation relies on PyTorch [59] for the core framework and PyTorch Geometric (PyG) [60] for graph-based operations. The number of stages and their feature dimensions are flexibly configured to align with the requirements of different downstream tasks, such as object detection and action recognition. To ensure efficient processing of our custom graph operations, parts of PyG's underlying code were modified. The training pipeline is managed with PyTorch Lightning on NVIDIA GPUs, and the model is optimized using the AdamW optimizer [61] with a OneCycle learning rate schedule [62].

### 4.3 Benchmark Comparison

#### 4.3.1 Object Detection

To comprehensively evaluate the effectiveness of EventMG on event-based object detection tasks, we compare it with several representative methods on two widely-used benchmark datasets: Gen1 and 1Mpx. The detailed results are presented in Table 1.

On the Gen1 dataset, EventMG demonstrates competitive performance, achieving the mAP of 53.7%. Notably, the model stands out for its compact model size; including the task head, it has only 1.96M parameters, making it very lean compared to similar methods. For instance, ERGO-12 [44] reaches 50.4% mAP with 59.6M parameters, whereas EventMG achieves higher accuracy with only 3.3% of its model size. Furthermore, the efficiency advantage of the model also extends to inference, where it runs significantly faster than high-performance models such as ERGO-12 (69.9 ms).

On the more challenging and higher-resolution 1Mpx dataset, EventMG remains competitive, achieving an mAP of 47.4%, which is comparable to the best-performing result from GET-T [9] (48.4%). This performance is achieved while maintaining a significant efficiency advantage, utilizing a small fraction of the parameters in contrast to GET-T's 21.9M.

In summary, EventMG matches or outperforms existing methods across datasets while significantly reducing model complexity and inference latency, demonstrating its effectiveness in learning and integrating spatiotemporal patterns from event streams.

#### 4.3.2 Action Recognition

Table 2: Comparison on THU$^{\text{E-ACT}}$-50-CHL dataset.

| Method | Params (M) | GFLOPs | Acc |
|---|---|---|---|
| HMAX SNN [67] | - | - | 0.327 |
| Motion-based SNN [39] | - | - | 0.473 |
| EV-ACT [53] | 21.3 | 29 | 0.585 |
| EventMamba [10] | 0.905 | 0.953 | **0.594** |
| **EventMG (Ours)** | **0.773** | **0.691** | 0.588 |

Table 3: Evaluation results on DVS Action dataset.

| Method | Params (M) | GFLOPs | Acc |
|---|---|---|---|
| STCA [68] | 15.13 | - | 0.792 |
| Motion-based SNN [39] | - | - | 0.781 |
| ST-EVNet [69] | 1.6 | - | 0.887 |
| TTPOINT [14] | **0.334** | 0.587 | 0.927 |
| EventMamba [10] | 0.905 | **0.476** | 0.891 |
| Two-stream SNN [70] | 9.17 | - | 0.917 |
| **EventMG (Ours)** | 0.773 | 0.531 | **0.933** |

We evaluate EventMG on event-based action recognition across four popular datasets, comparing it with leading methods. Results are shown in Tables 2, 3, and 4, with some baselines referenced from [10] and supplemented by original sources.

On the challenging THU$^{\text{E-ACT}}$-50-CHL dataset (Table 2), EventMG delivers strong performance. Compared to EV-ACT [53], it achieves higher accuracy with only 3.6% of the parameters and 2.3% of the GFLOPs. Although slightly behind EventMamba [10] in accuracy (0.588 vs. 0.594), EventMG requires fewer parameters (0.773M vs. 0.905M) and less computation (0.691 vs. 0.953 GFLOPs). These results confirm the efficiency of our proposed multilevel spatiotemporal modeling strategy.

On the DVS Action dataset (Table 3), EventMG achieves the highest accuracy among all methods at 93.3%, while maintaining a relatively low parameter count (0.773M) and the second-lowest computational cost (0.531 GFLOPs). These results highlight its balance between accuracy and efficiency. Although prior work [10] noted the risk of overfitting on small-scale datasets, EventMG mitigates this through the APN module, which enhances generalization to salient spatiotemporal patterns, ensuring robust and efficient performance.

For the larger HMDB51-DVS and UCF101-DVS datasets converted from conventional videos, EventMG demonstrates strong overall performance (Table 4). It achieves the lowest GFLOPs among all methods and keeps the parameter count below 1M—slightly above TTPOINT [14] but still far smaller than most advanced models. In terms of accuracy, while marginally below the best-reported results in [10], EventMG outperforms most existing approaches, including TTPOINT. This favorable trade-off is achieved without increasing internal feature dimensions, validating the effectiveness of the proposed multilevel spatiotemporal architecture in modeling complex dynamic scenes efficiently.

Table 4: Evaluation results on HMDB51-DVS and UCF101-DVS.

| Method | HMDB51-DVS | | | UCF101-DVS | | |
|---|---|---|---|---|---|---|
| | Params (M) | GFLOPs | Acc | Params (M) | GFLOPs | Acc |
| C3D [71] | 78.41 | 39.69 | 0.417 | 78.41 | 39.69 | 0.472 |
| I3D [72] | 12.37 | 30.11 | 0.466 | 12.37 | 30.11 | 0.635 |
| ResNet-34 [73] | 63.70 | 11.64 | 0.438 | - | - | - |
| ResNext-50 [74] | 26.05 | 6.46 | 0.394 | 26.05 | 6.46 | 0.602 |
| RG-CNN [55] | 3.86 | 12.39 | 0.515 | 6.95 | 12.46 | 0.632 |
| EVTC+I3D [75] | - | - | 0.704 | - | - | 0.791 |
| EventMix [76] | 12.63 | 1.99 | 0.703 | - | - | - |
| Event-LSTM [77] | - | - | - | 21.4 | - | 0.776 |
| TTPOINT [14] | **0.345** | 0.587 | 0.569 | **0.357** | 0.587 | 0.725 |
| EventMamba [10] | 0.916 | 0.953 | **0.864** | 3.28 | 3.722 | **0.979** |
| **EventMG (Ours)** | 0.773 | **0.531** | 0.712 | 0.773 | **0.531** | 0.796 |

## 4.4 Ablation Studies

This section presents ablation studies to assess the impact of key EventMG components. Starting from the full model, each module is removed individually and evaluated across two tasks (Table 5).

Table 5: **Ablation results on Gen1 and THU$^{\text{E-ACT}}$-50-CHL.** Gen1 corresponds to object detection, and THU$^{\text{E-ACT}}$-50-CHL to action recognition. Note: Parameter counts on Gen1 exclude the task head ($\sim$1.2M).

| Ablation Modules | | | Gen1 (Detection) | | | THU$^{\text{E-ACT}}$-50-CHL (Action) | | |
|---|---|---|---|---|---|---|---|---|
| Component Graph | APN | MSM | Params (M) | mAP | Time (ms) | Params (M) | GFLOPs | Acc |
| ✗ | ✗ | ✗ | 0.367 | 0.213 | 3.20 | 0.334 | 0.397 | 0.322 |
| ✗ | ✓ | ✓ | 0.639 | 0.447 | 6.95 | 0.632 | 0.608 | 0.491 |
| ✓ | ✗ | ✓ | 0.555 | 0.410 | 6.81 | 0.561 | 0.589 | 0.502 |
| ✓ | ✓ | ✗ | 0.630 | 0.424 | 5.26 | 0.685 | 0.585 | 0.486 |
| ✓ | ✓ | ✓ | 0.729 | 0.537 | 8.66 | 0.773 | 0.691 | 0.588 |

**Component Graph** models structural relationships among event clusters to provide macro-level semantics. Removing this module results in a significant accuracy drop in both tasks, highlighting its essential role in global structure modeling and cross-level feature fusion.

**SEST-APN (Adaptive Perturbation Network)** applies learnable coordinate perturbations to enhance robustness and focus on salient spatiotemporal patterns. Its removal reduces complexity but causes a sharp accuracy drop, confirming its value for dynamic adaptation and feature learning.

**SEST-MSM (Multi-directional Scanning Module)** captures spatiotemporal dependencies through multidirectional scanning. Excluding this module degrades performance in both tasks, demonstrating its importance in modeling diverse dynamic trajectories.

**Impact of Scanning Directions.** We conduct an ablation study on the scanning directions to validate our multidirectional design. For this study, we employ a configuration of forward scans along the $t$, $x$, and $y$ axes, which represents an effective balance of performance and efficiency. On the Gen1 dataset, this three-direction model achieves an mAP of 53.7%, showing a significant synergistic gain compared to unidirectional scans ($t$-axis only: 42.4%, $x$-axis only: 44.0%, and $y$-axis only: 43.8%).

## 5 Conclusion and Discussion

We propose EventMG, a Mamba-Graph collaborative architecture that systematically addresses the intrinsic challenges of event data. It leverages GNNs for local topology and an innovative SEST module to intelligently serialize graph information, enabling Mamba to model long-range dependencies with linear complexity while achieving a multi-level understanding of the data. The core contribution of EventMG lies in a novel design philosophy that seeks a delicate balance among representational power, computational efficiency, and respect for the data's intrinsic properties.

The limitations and future directions of EventMG warrant further exploration, including its generalization across diverse tasks, robustness under extreme conditions, and the challenges of hyperparameter tuning and interpretability inherent to its multi-module design. Future research should focus on exploring more efficient adaptive and interpretable methods to further refine this architectural paradigm.

## Acknowledgments and Disclosure of Funding

The authors declare no conflict of interest. This work was supported by the National Key Research and Development Program (No. 2024YFE0200703). The computations in this research were performed using the CFFF platform of Fudan University.

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

# A Preliminaries

## A.1 State Space Models

State Space Models (SSMs) [78] provide a classical framework for modeling dynamic systems by capturing the relationships between inputs and outputs through hidden states. These models describe the evolution of a system in terms of state variables that are not directly observed. Formally, an SSM is expressed as:

$$
\begin{aligned}
h'(t) &= Ah(t) + Bx(t), \\
y(t) &= Ch(t),
\end{aligned}
\tag{9}
$$

where $h(t) \in \mathbb{R}^N$ represents the hidden state, $x(t)$ is the input signal, and $y(t)$ is the output signal. The matrices $A$, $B$, and $C$ are system parameters that govern the dynamics of the system.

For discrete-time inputs, the Zero-Order Hold (ZOH) method [79] is commonly used to discretize the continuous-time model. The discretization of the system is given by:

$$
\begin{aligned}
\bar{A} &= \exp(\Delta A), \\
\bar{B} &= (\Delta A)^{-1}(\exp(\Delta A) - I) \cdot \Delta B,
\end{aligned}
\tag{10}
$$

where $\Delta$ denotes the time step and $\exp(\Delta A)$ represents the matrix exponential. The discretized system is then described by the following equations:

$$
\begin{aligned}
h_t &= \bar{A}h_{t-1} + \bar{B}x_t, \\
y_t &= Ch_t.
\end{aligned}
\tag{11}
$$

Although this formulation allows for modeling of discrete-time inputs, the assumption of linear time-invariance (LTI) restricts its flexibility. This limitation makes SSMs less adaptable to dynamic or non-uniform data patterns, where the system dynamics may change over time.

## A.2 Mamba: Linear-Time Sequence Modeling with Selective State Spaces

Mamba [25] extends the classical State Space Model framework by introducing dynamic, input-dependent parameters, allowing the system to adapt in real time to varying input sequences. The key difference lies in the system's ability to modify its parameters based on the current input, enabling a more flexible representation of dynamic systems. The formulation of Mamba is as follows:

$$
\begin{aligned}
h_t &= A(x_t)h_{t-1} + B(x_t)x_t, \\
y_t &= C(x_t)h_t,
\end{aligned}
\tag{12}
$$

where $A(x_t)$, $B(x_t)$, and $C(x_t)$ are dynamic, learnable parameters that depend on the input sequence. This flexibility allows Mamba to capture both long-range dependencies and local features, making it more adaptable to complex data patterns.

A distinctive feature of Mamba is its selective mechanism, which enables the model to decide which information should be propagated or discarded based on the current input token. This selective information flow improves computational efficiency and enhances the ability of the model to capture crucial dependencies in the data. By adjusting the influence of different parts of the input, Mamba is able to focus on the most relevant information at each time step, leading to more efficient learning and better generalization.

## A.3 Graph Neural Networks

Graph Neural Networks (GNNs) [80–84] are deep learning models tailored for graph-structured data, designed to extract high-order features from non-Euclidean domains by modeling interactions between nodes. Unlike traditional deep learning models such as convolutional neural networks (CNNs) and recurrent neural networks (RNNs), which primarily target Euclidean data with regular structures (e.g., images and text), GNNs are specifically built for scenarios where data is inherently represented as graphs. Examples include social networks, molecular structures, and recommendation systems, where connections between entities are complex and dynamically evolving.

The core mechanism of GNNs is *message passing*, which iteratively aggregates information from neighboring nodes to update node representations. This approach enables GNNs to capture global

structural patterns and solve tasks such as node classification, link prediction, and graph classification, establishing their significance in graph representation learning.

Given a graph $G = (V, E)$, where $V = \{v_1, v_2, \ldots, v_N\}$ is the set of nodes and $E \subseteq V \times V$ is the set of edges, each node $v \in V$ is associated with an initial feature vector $\mathbf{h}_v^{(0)} \in \mathbb{R}^d$, and edges $e_{uv} \in E$ may carry feature vectors $\mathbf{h}_{uv} \in \mathbb{R}^k$. The neighborhood of a node $v$ is denoted as $\mathcal{N}(v) = \{u \mid (u, v) \in E\}$. The update process at each layer in a GNN can be expressed as:

$$\mathbf{h}_v^{(l+1)} = \phi_{\text{update}} \left( \mathbf{h}_v^{(l)}, \psi_{\text{aggregate}} \left( \{ \mathbf{h}_u^{(l)} \mid u \in \mathcal{N}(v) \} \right) \right), \tag{13}$$

where $\mathbf{h}_v^{(l)}$ is the representation of node $v$ at layer $l$, $\psi_{\text{aggregate}}$ aggregates information from neighboring nodes (e.g., sum, mean, max), and $\phi_{\text{update}}$ updates the node representation using this aggregated information.

Early implementations of GNNs were introduced by Scarselli et al. [80], forming the foundation for subsequent advancements. Over time, specialized variants have been developed to enhance the expressiveness and efficiency of GNNs in different applications.

**Graph Convolutional Network (GCN):** The GCN model [81] simplifies the message-passing process by using spectral graph convolutions. It enhances local feature propagation through normalized adjacency matrices, making it efficient for smoothing features across neighbors. Its layer-wise update is defined as:

$$\mathbf{H}^{(l+1)} = \sigma \left( \tilde{\mathbf{D}}^{-\frac{1}{2}} \tilde{\mathbf{A}} \tilde{\mathbf{D}}^{-\frac{1}{2}} \mathbf{H}^{(l)} \mathbf{W}^{(l)} \right), \tag{14}$$

where $\tilde{\mathbf{A}} = \mathbf{A} + \mathbf{I}$ is the adjacency matrix with self-loops, $\tilde{\mathbf{D}}$ is the degree matrix, and $\mathbf{W}^{(l)} \in \mathbb{R}^{d^{(l)} \times d^{(l)}}$ is a learnable weight matrix.

**Graph Attention Network (GAT)** GAT [82] introduces an attention mechanism, allowing nodes to dynamically weigh the importance of their neighbors during aggregation. This enables GAT to capture heterogeneous relationships within the graph effectively. The update rule is:

$$\mathbf{h}_v^{(l+1)} = \sigma \left( \sum_{u \in \mathcal{N}(v) \cup \{v\}} \alpha_{vu}^{(l)} \mathbf{W}^{(l)} \mathbf{h}_u^{(l)} \right), \tag{15}$$

where $\alpha_{vu}^{(l)}$ are attention coefficients computed by an attention mechanism (typically a shared linear transformation followed by a LeakyReLU activation and normalized via Softmax across $u$), and $\mathbf{W}^{(l)}$ is a learnable weight matrix for transforming node features. Note the inclusion of $v$ in its own neighborhood for self-attention.

**Graph Isomorphism Network (GIN)** GIN [83] is designed to achieve maximum discriminative power, theoretically matching the expressiveness of the Weisfeiler-Lehman (WL) graph isomorphism test. It optimizes aggregation and update functions, often using a Multi-Layer Perceptron (MLP). A common update rule is:

$$\mathbf{h}_v^{(l+1)} = \text{MLP}^{(l)} \left( (1 + \epsilon^{(l)}) \cdot \mathbf{h}_v^{(l)} + \sum_{u \in \mathcal{N}(v)} \mathbf{h}_u^{(l)} \right), \tag{16}$$

where $\text{MLP}^{(l)}$ is a layer-specific MLP, and $\epsilon^{(l)}$ is either a learnable parameter or a fixed scalar that adjusts the weight of the central node's feature.

**GraphSAGE** GraphSAGE (Graph SAmple and aggreGatE) [84] focuses on inductive representation learning, enabling GNNs to generalize to unseen nodes or entirely new graphs. Instead of aggregating features from all neighbors, it typically samples a fixed-size neighborhood for each node and applies an aggregation function. A general update step can be written as:

$$\begin{aligned} \mathbf{h}_{\mathcal{N}(v)}^{(l)} &= \text{AGGREGATE}^{(l)} \left( \{ \mathbf{h}_u^{(l-1)} \mid u \in \mathcal{N}_{\text{sampled}}(v) \} \right) \\ \mathbf{h}_v^{(l)} &= \sigma \left( \mathbf{W}^{(l)} \cdot \text{CONCAT}(\mathbf{h}_v^{(l-1)}, \mathbf{h}_{\mathcal{N}(v)}^{(l)}) + \mathbf{b}^{(l)} \right), \end{aligned} \tag{17}$$

where $\mathcal{N}_{\text{sampled}}(v)$ is the sampled neighborhood of $v$, $\text{AGGREGATE}^{(l)}$ can be functions like mean, LSTM, or pooling, CONCAT denotes concatenation, and $\mathbf{W}^{(l)}$ and $\mathbf{b}^{(l)}$ are learnable parameters. GraphSAGE is particularly effective for inductive tasks.

Each of these models enhances GNN capabilities through distinct technical innovations, achieving great performance in tasks like node and graph classification.

# B    Discussion on Theoretical Limitations

This section provides an objective discussion on the inherent theoretical limitations of the foundational modules our model relies on (GNNs and SSMs), as well as the analysis of the interpretability of our proposed APN module. Understanding these limitations is key to clarifying the trade-offs and motivations behind our EventMG hybrid architecture.

## B.1    Limitations of Graph Neural Networks

GNNs based on message passing are powerful for processing structured data, but they also face several inherent theoretical challenges:

**Over-smoothing:** In message-passing GNNs, the nodes' features are formed by aggregating information from their neighborhoods. As the network deepens, the receptive field of each node expands and begins to significantly overlap with others. This causes the representations of different nodes to become homogenous, ultimately losing their distinctiveness and discriminative power. Consequently, modeling long-range dependencies by simply stacking GNN layers is extremely difficult, as the deepening of the network is the very cause of this feature smoothing.

**Scalability Limitations:** When processing large-scale graphs or those with high-degree nodes, the computational and memory costs of GNNs can grow exponentially with depth, a phenomenon known as "neighbor explosion." This poses significant challenges to model scalability.

These limitations, particularly the over-smoothing problem, highlight the inadequacy of solely stacking GNN layers for long-range dependency modeling and motivate the integration with other paradigms, such as SSMs.

## B.2    Limitations of State Space Models

SSMs, as represented by Mamba, demonstrate excellent capabilities in capturing long-range dependencies with linear complexity. However, their nature as sequence models also introduces inherent limitations:

**Topology-agnostic Nature:** The fundamental design of an SSM is to process one-dimensional, ordered sequences, where its core function is to understand "before-and-after" relationships. However, the structure of high-dimensional data such as graphs and images is topological and lacks a single, natural sequential order. Therefore, before applying SSM, the data must be "flattened" into a 1D sequence through a scanning process. This step is a performance bottleneck: a poor serialization strategy can destroy the original local topology, preventing the SSM from learning meaningful patterns. Consequently, an SSM's effectiveness is highly dependent on the design of a "smart serializer" that can effectively encode high-dimensional structural information into a 1D sequence.

**Inherent Unidirectionality:** Standard SSMs, including Mamba, are causal models that only process unidirectional information flow. This is a limitation for many perception tasks that require global context, unlike autoregressive generation tasks. Although this can be compensated for by using bidirectional scanning, it typically requires additional computational and design complexity.

These limitations, especially the topology-agnostic nature, highlight the necessity of designing a "smart serialization" front-end for SSMs that is aware of multidimensional structures.

## B.3    Limitations in the Interpretability of APN

We had planned to provide a more intuitive validation for the APN module's Effective Coordinate Space hypothesis by visualizing the coordinate perturbation values $(\Delta x, \Delta y, \Delta t)$. Our initial assump-

tion was that these perturbations might exhibit an easily interpretable pattern, such as larger spatial perturbations occurring at object edges to achieve a separation between foreground and background.

However, after conducting rigorous and comprehensive experiments on multiple samples, we found that these perturbation values exhibit a high degree of non-linearity and sample dependency. Although the expected trends were observed in a minority of samples, in most cases, the distribution of perturbations presented complex patterns that are difficult to interpret intuitively. We attribute this primarily to the following reasons:

**The Non-linear Nature of Optimization:** The perturbations learned by APN are not meant to align with human intuition. Instead, they are driven by the final task loss to create optimal 1D sequences for the subsequent MSM and Mamba modules. This optimization process is highly non-linear, and its final solution may be a complex yet effective mapping that is not directly interpretable by humans.

**Collective vs. Individual Effect:** The effectiveness of APN likely stems from the collective effect of all node perturbations, rather than relying on a few individual nodes producing large, intuitive offsets. While the perturbation of a single node may have limited significance, the aggregate effect of all perturbations collectively forms a coordinate space that is more advantageous for scanning.

We believe that presenting only a few well-behaved samples carries the risk of cherry-picking evidence and may not objectively reflect the general behavior of APN in all cases. This essentially touches upon the challenge we have already noted in the main Discussion section: the interpretability of complex models' internal mechanisms. Based on this consideration, we have decided to omit this visual analysis from the current version and plan to explore this more challenging topic as a dedicated direction for mechanistic research in future work.

## C    Downstream Task

### C.1    Graph-to-Vision Pyramid

The Graph-to-Vision Pyramid (GVP) module is specifically designed for downstream vision tasks, with the core objective of converting the sparse, unstructured graph node features from upstream modules into the dense, multi-scale pyramidal feature maps favored by vision applications.

The core of this transformation is an efficient, index-based Sparse-to-Dense Mapping strategy. Specifically, for each graph node feature $\mathbf{h}_i$ that fuses micro- and macro-level information, we place it directly onto a 2D feature grid at its corresponding original pixel coordinate $(x_i, y_i)$. This process initially forms a sparse feature map. Subsequently, we process this sparse grid with standard convolutional layers to densify and convert it into a dense base feature map. From this base map, multi-layer projection operations are used to generate feature maps at different resolutions, $\{\mathbf{F}_1, \mathbf{F}_2, \ldots, \mathbf{F}_K\}$, which collectively form the visual pyramid.

It is worth noting that while this process does not explicitly preserve the graph's edge structure, the key semantic information is effectively embedded into the final multi-scale visual features because the input node features $\mathbf{h}_i$ have already been deeply encoded with rich spatiotemporal and structural relationships by the upstream modules. This mapping strategy, being based entirely on indexing operations, is highly parallelizable and therefore extremely efficient for GPU execution.

For non-visual tasks or those that only require a global feature representation, the GVP module can be replaced with other more suitable pooling or processing techniques. This modular design ensures the flexibility and adaptability of our overall framework to a diverse range of tasks.

### C.2    Detection Head: RT-DETR

Our EventMG employs RT-DETR [85] as its detection head, with a parameter size of approximately 1.20M. RT-DETR (Real-Time Detection Transformer) is the first end-to-end real-time detection model, designed to accelerate the DETR (Detection Transformer) framework while enhancing query selection for improved detection accuracy. By addressing computational bottlenecks in traditional Transformer-based object detection, RT-DETR significantly enhances both efficiency and accuracy, making it a practical choice for real-time vision applications.

**1. Architecture and Key Enhancements** RT-DETR is composed of three principal components. First, the **backbone** extracts multi-scale features from the input image, serving as the foundation for subsequent processing. Second, the **Efficient Hybrid Encoder** improves computational efficiency by decoupling intra-scale feature interactions from cross-scale feature fusion, effectively reducing redundancy while maintaining rich contextual information. Third, the **Transformer Decoder** employs iterative refinement to optimize object queries using auxiliary prediction heads.

A key innovation in RT-DETR is the **Uncertainty-Minimal Query Selection** mechanism, which refines target queries by reducing ambiguity, leading to greater robustness and a lower false positive rate. This mechanism ensures that the most relevant object representations are selected during detection, improving precision in complex scenes.

**2. Loss Function** RT-DETR follows an end-to-end training approach similar to DETR, incorporating a dynamic matching strategy for target assignment. The loss function is formulated as:

$$\mathcal{L} = \lambda_{\text{cls}}\mathcal{L}_{\text{cls}} + \lambda_{\text{box}}\mathcal{L}_{\text{box}} + \lambda_{\text{giou}}\mathcal{L}_{\text{giou}}, \tag{18}$$

where $\mathcal{L}_{\text{cls}}$ represents the classification loss, implemented using Focal Loss to improve the handling of imbalanced class distributions. The term $\mathcal{L}_{\text{box}}$ is the L1 loss applied to bounding box regression, ensuring precise localization of detected objects. Additionally, $\mathcal{L}_{\text{giou}}$ employs Generalized IoU (GIoU) loss, which enhances bounding box alignment and optimizes spatial consistency between predicted and ground-truth boxes.

**3. Performance and Efficiency** RT-DETR achieves state-of-the-art results on the COCO dataset, demonstrating a strong balance between detection accuracy and inference speed. RT-DETR-R50 achieves 53.1% mAP while running at 108 FPS on an NVIDIA T4 GPU, while RT-DETR-R101 reaches 54.3% mAP at 74 FPS. These results outperform many existing YOLO-based models, positioning RT-DETR as a highly effective choice for real-time detection scenarios.

Compared to conventional Transformer-based detectors, RT-DETR significantly reduces computational overhead while maintaining end-to-end training benefits. Its capability to efficiently process complex object scenes makes it particularly well-suited for event-based vision applications within the EventMG framework.

### C.3 Detection Head: YOLOX

YOLOX [86] is included in our study as an optional alternative detection head. Although detection results based on YOLOX are not reported in this work, it remains one of the widely adopted detection heads in the field. As an advanced variant of the YOLO series, YOLOX enhances detection accuracy while maintaining computational efficiency. It improves upon YOLOv3 and YOLOv5 by introducing an anchor-free paradigm, a decoupled detection head, and the SimOTA target assignment strategy, effectively addressing the limitations of anchor dependency and suboptimal target assignment in earlier YOLO models.

**1. Architecture and Key Enhancements** The YOLOX architecture comprises four main components. The backbone, typically based on DarkNet53 or CSPNet, extracts hierarchical feature representations from the input image. The Feature Pyramid Network (FPN) and Path Aggregation Network (PAN) refine multi-scale feature fusion, significantly improving the detection of small objects. The decoupled detection head independently processes classification and regression tasks, reducing task conflicts and improving both convergence speed and accuracy. Lastly, the SimOTA target assignment strategy dynamically assigns positive samples using optimal transport theory, leading to superior target matching and robustness.

Three major improvements distinguish YOLOX from its predecessors. First, the anchor-free detection approach eliminates reliance on predefined anchor boxes, simplifying training and reducing the need for hyperparameter tuning. Second, the decoupled detection head enables separate processing for classification and localization, leading to higher precision in object detection. Third, the SimOTA target assignment mechanism enhances label assignment flexibility, particularly benefiting the detection of occluded and small objects.

**2. Loss Function**  The loss function of YOLOX is formulated as follows:

$$\mathcal{L} = \mathcal{L}_{\text{cls}} + \lambda_{\text{reg}}\mathcal{L}_{\text{reg}} + \lambda_{\text{iou}}\mathcal{L}_{\text{iou}}, \tag{19}$$

where: - $\mathcal{L}_{\text{cls}}$ represents the classification loss (Cross Entropy Loss). - $\mathcal{L}_{\text{reg}}$ denotes the bounding box regression loss (L1 loss). - $\mathcal{L}_{\text{iou}}$ is the IoU-based loss, optimizing bounding box localization accuracy.

**3. Performance and Efficiency**  YOLOX achieves state-of-the-art performance on the COCO dataset. Specifically, YOLOX-L attains 50.0% AP on 640×640 input resolution, outperforming YOLOv5-L (48.2% AP). Additionally, its lightweight variants, YOLOX-Tiny and YOLOX-Nano, achieve 32.8% and 25.3% AP, respectively, making them highly suitable for real-time and resource-constrained applications.

Compared to RT-DETR, YOLOX's anchor-free framework facilitates faster inference and lower computational overhead. However, it may be less suited for fully end-to-end detection tasks due to its separate processing pipeline. Within the EventMG framework, YOLOX is employed as an alternative detection head, offering flexibility for different event-based vision applications, particularly those requiring real-time performance.

## D  Hyperparameter Sensitivity Analysis

This section presents a sensitivity analysis of several key hyperparameters in EventMG to validate the robustness of our model design and the rationale for our parameter choices. All experiments are conducted on the Gen1 dataset, with mAP used as the performance metric.

Table 6: Hyperparameter sensitivity analysis. All experiments are conducted on the Gen1 dataset, with mAP as the metric. Bold values indicate the default settings used in our paper.

| $N$ | | $n_{\text{min}}$ | | $\delta_{\text{st}}$ | |
|---|---|---|---|---|---|
| Value | mAP | Value | mAP | Value | mAP |
| 10,000 | 0.471 | 50 | 0.529 | 0.01 | 0.498 |
| **20,000** | **0.537** | **100** | **0.537** | 0.02 | 0.531 |
| 30,000 | 0.549 | 150 | 0.534 | **0.05** | **0.537** |
| | | 200 | 0.522 | 0.10 | 0.533 |
| | | | | 0.15 | 0.534 |

### D.1  Impact of Event Count

In our method, we employ a strategy of using a fixed number of events, $N$ (rather than a fixed time window), to segment the event stream. This design effectively constitutes an Activity-driven Dynamic Frame Rate: in scenes with dense events, the time required to acquire $N$ events is short, equivalent to a 'high frame rate'; conversely, in sparse scenes, the time taken is longer, equivalent to a 'low frame rate'. This mechanism allows the model to avoid processing information-less 'empty windows,' thereby enabling efficient use of computational resources.

To investigate the impact of the value of N on performance, we conducted a sensitivity analysis, with the results shown in Table 6. The analysis clearly shows that when N is halved from 20,000 to 10,000, the model's performance drops significantly (-12.3%) due to insufficient input information and an inability to capture the complete spatiotemporal structure. Conversely, increasing N by 50% to 30,000 yields only a marginal gain in performance (+2.2%). Therefore, N=20,000 is selected as an effective balance point between accuracy and efficiency for the model.

### D.2  Impact of Minimum Component Size

The parameter $n_{min}$ is used to filter out excessively small event clusters when constructing the macro-level component graph. In our design, $n_{min}$ acts merely as a coarse-grained noise filter rather than a precise object segmenter. Its effectiveness is based on a key observation: event clusters generated by the motion of real objects are typically much larger than those formed by random noise.

More importantly, our subsequent macro-micro fusion design provides a degree of fault tolerance. The final refined predictions still rely on micro-level features, with macro-level information serving only as contextual guidance. This means that even if the filtering by $n_{min}$ is imperfect (e.g., one component containing multiple objects, or multiple components corresponding to a single object), the model can still distinguish them based on micro-level features, which greatly reduces the system's sensitivity to $n_{min}$.

The quantitative analysis results, as shown in Table 6, confirm the robustness of our design. Adjusting $n_{min}$ within a four-fold range (50 to 200) results in largely stable model performance, proving that the overall architecture is not sensitive to this hyperparameter.

### D.3 Impact of Spatiotemporal Distance Threshold

The threshold $\delta_{st}$ is used to define the neighborhood range of nodes when constructing the initial event graph. Our sensitivity analysis on this parameter (see Table 6) reveals a key phenomenon of asymmetric sensitivity.

On one hand, when $\delta_{st}$ is too small (e.g., 0.01), model performance drops sharply. This is because event data is not uniformly distributed in spacetime but rather clusters along contours and edges generated by object motion. An excessively small threshold fails to connect these physically close-related events, thereby disrupting the graph's fundamental topology.

On the other hand, once $\delta_{st}$ reaches a reasonable value capable of capturing the local neighborhood (e.g., $\geq 0.02$), the model's performance enters a broad and stable operating range. This robustness to larger threshold values is due to two safeguard mechanisms in our design: 1) The hard constraint of the maximum number of neighbors, $k$, prevents nodes from connecting to too many neighbors even with a large threshold; 2) The micro-macro fusion feedback architecture has strong fault tolerance, which can mitigate the noise effects of redundant connections.

This analysis indicates that our model does not rely on a fragile optimal value for $\delta_{st}$. Although adaptive threshold schemes, such as [87], represent a more advanced direction in the field, they are often accompanied by higher computational costs. How to strike a balance between the flexibility of adaptive methods and the simple efficiency of our current strategy is a topic worthy of future exploration.

