# OpenReview forum: "EventMG: Efficient Multilevel Mamba-Graph Learning for Spatiotemporal Event Representation"
_NeurIPS.cc/2025/Conference — NeurIPS 2025 poster_

### Official Review · Reviewer_Xb3d · 2025-07-01

**Clarity:** 1
**Significance:** 2
**Originality:** 2
**Rating:** 4
**Confidence:** 3

**Summary:**

The paper introduces a hybrid network building on state space models and GNNs to process events. They evaluate the method on object detection and action recognition.

**Questions:**

- Did you perform any subsampling of events before building the graph (eq. 1)?
- How did you find specific parameters like the scaling coefficient and threshold delta_st?

**Ethical Concerns:**

["NO or VERY MINOR ethics concerns only"]

**Final Justification:**

I thank the authors for their extensive reply to my concerns. I updated my rating to borderline accept.

**Limitations:**

The authors mention some limitations in the discussion section, especially a discussion of theoretical limitations, e.g. of GNNs or SSMs, could strengthen the paper.

**Paper Formatting Concerns:**

-

**Quality:**

2

**Strengths And Weaknesses:**

**Strength**
- Addresses a challenge in event vision to build neural networks that use the sparse nature of the event data.
- The empirical results for object detection and action recognition in combination with a low parameter count are promising.

**Weaknesses**
- The presentation of the network is a bit confusing. I think the overall workflow and how SEST is embedded in it, could be explained better.
- There are limited ablations. Tab. 5 shows ablations for the main modules, but there seem to be a lot of hyperparameters (e.g., component size threshold).

---

> ### Author Response · Authors · 2025-08-01
> **Rebuttal to Reviewer Xb3d (Part 1 of 2)**
>
> Dear Reviewer Xb3d,
>
> We sincerely apologize, but due to irresolvable technical issues at the submission deadline, we were unable to successfully submit our complete Rebuttal. Therefore, we are submitting our formal response to your valuable comments directly in this comment section.
>
> Thank you for your understanding and thoughtful review.
>
> Note: due to this section's 5,000-character limit (compared to the original 10,000), we split our response into two parts. This is **Part 1**.
>
> ---
>
> **W1:** We sincerely apologize for the lack of clarity regarding the overall workflow and the integration of the SEST module, and we thank you for highlighting this issue. We have substantially revised the relevant sections for clarity.
>
> To address your concern about the "overall workflow," we now provide an explicit overview of our model's four core stages at the beginning of the methodology section:
>
> 1. **Basic Graph Construction**: Captures local spatiotemporal topology.
> 2. **Micro-level Modeling**: Extracts long-range dynamics via the SEST and EventsMamba.
> 3. **Macro-level Modeling**: Distills global context using a Component Graph.
> 4. **Feedback and Fusion**: Infuses global context into micro-level features to enhance representations.
>
> The confusion you identified—how SEST is embedded—arises from our original conflation of serialization and feature extraction in the micro-level stage. In the revised version, we clearly separate these roles:
>
> SEST acts as an intelligent serializer, converting the event graph into a set of directional sequences $\mathcal{S}$ for the next stage:
> $$
> \mathcal{S} = \{ S_k = \text{Scan}(V_i, d_k) \mid d_k \in D \},
> $$
>
> where the set of directions D={t+,t-,x+,x-,y+,y-}. EventsMamba then performs feature extraction over $\mathcal{S}$:
> $$
> h_v = \underset{S_k \in \mathcal{S}}{\text{Aggr}} \left( \text{Mamba}(S_k) \right)
> $$
>
> To reinforce this clarification, we have updated the methodological description, corrected Equation (4), and comprehensively revised Figure 1 and its caption. We believe these revisions fully resolve the ambiguity and greatly improve the manuscript’s clarity. Thank you again for your valuable feedback.
>
>
>
> **W2:** Thank you for your valuable suggestions regarding the ablation studies and hyperparameter sensitivity. We fully agree that in-depth validation is essential to demonstrate the model's effectiveness.
>
> 1. **Regarding the Existing Ablation Studies**
> As shown in Table 5, our ablation study targets the verification of core architectural components. We independently remove the macro-level module (Component Graph) and the two main micro-level innovations (APN and MSM). The results clearly show that each of these modules plays an essential role in the overall performance.
>
> 2. **On Robustness to the Hyperparameter $n_{\text{min}}$**
> We acknowledge that, beyond the modules themselves, the setting of key hyperparameters also deserves discussion. Regarding the component size threshold you mentioned, our core design philosophy was to build a system that is inherently robust and not sensitive to its precise value. This robustness primarily stems from two aspects:
>
> - **Data-Driven Intrinsic Stability:**
>   Event data is not uniformly distributed; it naturally forms highly concentrated and dense clusters around the edges and textures of objects. Consequently, after the component graph is constructed, the sizes of components representing real objects are typically orders of magnitude larger than those formed by sporadic noise. This inherent skewed distribution ensures that the set of identified primary components remains highly stable across a wide range of $n_{\text{min}}$ thresholds.
>
> - **Architectural Fault Tolerance:**
>   Our macro-level module is designed to learn the relationships between components. Even if a suboptimal $n_{\text{min}}$ setting occasionally filters out a small but useful part of an object, the subsequent macro-level model can still infer the complete scene context from the interactions of the surrounding, more significant components. This largely compensates for and corrects the information loss caused by low-level local filtering.
>
> Given this inherent data property and architectural resilience, the model's performance is largely insensitive to the exact value of $n_{\text{min}}$. Therefore, in our experiments, we prioritized the ablation analysis of the core architectural modules.
>
> Nonetheless, we appreciate your suggestion and have included a detailed discussion of this robustness, along with our design rationale, in the appendix of the revised manuscript.
>
> ---
>
> (Due to length, the response continues in the **next** comment. This is Part 1 of 2.)

---

> ### Author Response · Authors · 2025-08-01
> **Rebuttal to Reviewer Xb3d (Part 2 of 2)**
>
> (Continuation of the response from the **previous** comment. This is **Part 2** of 2.)
>
> ---
>
> **Q1:** Thank you for your question; this is a critical point regarding our input data processing. The term "subsampling" has two interpretations here, which we clarify:
>
> 1. **Regarding Downsampling within a Single Processing Batch**
>    We do **not** perform any lossy downsampling (e.g., random sampling or voxel filtering) on an accumulated slice of events before constructing the graph. We aim to retain full raw event information, allowing the model to learn from rich details directly.
>    The only filtering step occurs **later** in the macro-level modeling, where we denoise by removing extremely small connected components. This is a guided, topology-based screening—not an indiscriminate, upfront downsampling.
>
> 2. **Regarding the Method of "Sampling" a Processing Batch from the Event Stream**
>    If *"subsampling"* refers to how we *extract* a batch from the continuous event stream to construct the graph (Eq. 1), then we employ a **Fixed-count Accumulation** strategy.
>    Specifically, we use a sliding window with a fixed number of events (e.g., N = 20,000) to slice the event stream into batches. This method segments a continuous stream into manageable batches that adapt to scene activity, ensure computational control, and maintain stable input size.
>
>    This accumulation method is widely used in event-based vision [1, 2].
>
> We apologize for not having clearly articulated this data processing pipeline in our original manuscript and have added a detailed supplementary explanation in the methodology section of our revised version.
>
> [1] Rethinking efficient and effective point-based networks for event camera classification and regression. TPAMI2025
> [2] Event-based vision: A survey. TPAMI2020
>
>
> **Q2:** Thank you for your question regarding our experimental details. The selection of these two parameters was indeed crucial to our model's performance. We adopted mature practices and empirically validated them.
>
> 1. **Regarding the Scaling Coefficient**
>    In our graph construction process, to ensure a fair comparison with related works, we directly adopted the setting from [3]. This approach allows us to validate the effectiveness of our model's other innovative modules on a recognized and mature foundation, while avoiding the introduction of excessive custom variables that could interfere with the experimental results.
>
> 2. **Regarding the Spatiotemporal Neighborhood Threshold (δₛₜ)**
>    For the key parameter in our graph construction—the spatiotemporal neighborhood threshold δₛₜ—we employed a **systematic, data-driven optimization process** to ensure its selection was both theoretically grounded and empirically optimal.
>    Before conducting the parameter search, we performed a **data pre-analysis** on the training set. By statistically analyzing:
>    - the average spatial distance to the 32 nearest neighbors (k = 32), and
>    - the distribution of timestamp differences between neighbors,
>
>    This informed a compact parameter search range. Then, on a strictly held-out validation set, we systematically evaluated each δₛₜ value, using final task performance (mAP) as the optimization target.
>
>  [3] AEGNN: Asynchronous Event-based Graph Neural Networks. CVPR 2022
>
>
> **Limitations:**
> Thank you very much for this highly constructive suggestion. We agree that discussing GNN and SSM limitations would enhance the paper's depth and clarify our hybrid design motivation.
>
> Following your valuable suggestion, we will add a **subsection in the appendix** of our revised manuscript to specifically discuss these theoretical limitations and how our design attempts to mitigate them. The main arguments we have added include:
>
> - **On the limitations of GNNs**
>   We discuss the *over-smoothing problem*, which is prevalent in standard message-passing GNNs when capturing long-range dependencies. This directly explains why we chose **not to simply stack numerous GNN layers**, but instead introduced an SSM as an alternative long-range modeling solution.
>
> - **On the limitations of SSMs (Mamba)**
>   We point out the *topology-agnostic nature* of SSMs as one-dimensional sequence models. Their performance is highly dependent on how high-dimensional spatiotemporal data is **serialized**. This precisely highlights the necessity of the **SEST module** in our method—it acts as an *"intelligent serializer,"* providing Mamba with **optimized, multi-directional sequence inputs**, thereby effectively encoding topological information into one-dimensional sequences.
>
> We believe this addition will enable readers to more comprehensively understand the trade-offs behind our design choices and to more clearly see the positioning of our work. Thank you again for your valuable feedback; it has helped us significantly improve the completeness of our paper.

---

> > ### Comment · Area_Chair_1GwA · 2025-08-05
> >
> > The authors had some technical difficulties and have uploaded rebuttals as comments. Please still read these and respond appropriately. Current ratings are borderline leaning reject. Please discuss with authors and reviews to come to a consensus between reviewers.

---

> > ### Author Response · Authors · 2025-08-08
> > **Thank You and Follow-up on Your Review**
> >
> > **Dear Reviewer Xb3d,**
> >
> > Thank you very much for the time and valuable feedback you provided during the initial review of our paper.
> >
> > In our previous response, we have carefully addressed each of your comments and revised the manuscript accordingly. We truly value your input and hope that our revisions have effectively addressed your concerns.
> >
> > As the Reviewer–Author Discussion phase draws to a close, we would sincerely appreciate it if you could let us know whether our responses and modifications have resolved your concerns, or if there remain any points requiring further clarification.
> >
> > Thank you again for your time and thoughtful guidance!
> >
> > Best regards,
> > *The Authors of Submission 27341*

---

### Official Review · Reviewer_NxX4 · 2025-07-02

**Clarity:** 3
**Significance:** 3
**Originality:** 3
**Rating:** 4
**Confidence:** 3

**Summary:**

This paper introduces EventMG, a lightweight, multilevel Mamba-Graph architecture designed for efficient spatiotemporal representation of event camera data. It combines a micro-level Mamba-based State Space Model, enhanced by a Spatiotemporal-aware Event Scanning Technology (SEST), with a macro-level graph-based representation of event clusters. Through a fusion of long-range event dependencies and local-global structural cues, EventMG effectively models the asynchronous, sparse nature of event data and achieves strong performance on object detection and action recognition tasks.

**Questions:**

1. Can the Multidirectional Scanning Module (MSM) be considered a form of data augmentation that effectively increases the diversity of training data? The MSM appears to contribute significantly to performance improvements, as evidenced in Table 5. I’m also curious whether similar scanning strategies are commonly adopted in related methods. If my understanding is incorrect, I would appreciate any clarification.

**Ethical Concerns:**

["NO or VERY MINOR ethics concerns only"]

**Final Justification:**

Most of my concerns have been addressed through the discussion. I’m leaning towards raising the score accordingly.

**Limitations:**

See weaknesses. My primary concern lies in the necessity of each specifically designed component within the overall complex network architecture. Given that the reported experimental improvements are relatively modest, a more in-depth analysis or justification of the full design would be valuable. I'm willing to re-evaluate this paper based on the author's feedback.

**Paper Formatting Concerns:**

No formatting concerns.

**Quality:**

3

**Strengths And Weaknesses:**

Strengths:
1. The paper clearly articulates its motivation to tackle key issues in prior approaches, such as spatiotemporal precision, representational adequacy, and computational overhead. These concerns are well-founded and convincingly presented.
2. The manuscript is well-organized and written in a clear, accessible manner, making it easy to follow.

Weaknesses:
1. The overall architecture appears somewhat complex. Although the rationale behind each network component is explained in the respective sections, the paper would benefit from additional experimental validation, such as ablation studies that replace specific modules with alternative designs or visualizations that highlight the contributions of tailored components. This would provide more intuitive insights beyond the quantitative results reported in Table 5.
2. While Table 5 shows a significant reduction in model parameters, this does not seem to translate into substantial practical speedup, for example, when compared to EGSST-B without the SSM module. This might be due to the sophisticated architecture being less amenable to GPU acceleration. Additionally, reporting FLOPs alongside inference time would offer a more comprehensive assessment of the model’s computational efficiency.

---

> ### Author Response · Authors · 2025-08-01
> **Rebuttal to Reviewer NxX4 (Part 1 of 2)**
>
> Dear Reviewer NxX4,
>
> We sincerely apologize, but due to irresolvable technical issues at the submission deadline, we were unable to successfully submit our complete Rebuttal. Therefore, we are submitting our formal response to your valuable comments directly in this comment section.
>
> Thank you for your understanding and thoughtful review.
>
> Note: due to this section's 5,000-character limit (compared to the original 10,000), we split our response into two parts. This is **Part 1**.
>
> ---
>
> **W1:** Thank you for your thoughtful suggestion. In response, we have incorporated several new sensitivity analyses of key hyperparameters in the revised manuscript, guided by your and other reviewers' valuable feedback.
>
> Specifically, as noted in our responses to reviewers 1mAU-W6 and rxhc-Q3, we have added quantitative analyses on the macro-level component threshold \( n_{\text{min}} \) and the event slice size \( N \). These results demonstrate the model's robustness to key hyperparameters and validate the rationale behind our default settings. We believe these additions directly address your core concern regarding the need for further empirical validation.
>
> While visualization is ideal for interpretability, we found it limited in this context. Spatial perturbations \( (\Delta x, \Delta y) \) in 3D spatiotemporal point clouds are visually subtle, and the more critical temporal adjustments \( (\Delta t) \) cannot be effectively illustrated in static images. Consequently, the clarity and persuasiveness of visualizations are inherently constrained.
>
> To offer a more intuitive understanding of our architecture's collaborative design, we present the following conceptual analogy:
>
> - GNN (local processing) acts like individual section players in an orchestra, aligning with their nearby peers to ensure local harmony.
> - Mamba (long-range modeling) serves as the recurring musical theme, capturing the structure and echoes between the beginning and end of the sequence.
> - The macro-level module functions as the conductor, maintaining the overarching narrative and emotional progression.
> - The final feedback mechanism resembles the conductor's precise cue to a percussionist—targeted, globally informed, and impactful.
>
> Together, these new quantitative studies and conceptual illustration aim to provide a more comprehensive and intuitive justification for the necessity and synergy of each architectural component. The corresponding experiments and discussions have been incorporated into the revised manuscript. Once again, we thank you for your insightful feedback, which significantly improved the rigor of our work.
>
>
> **W2:** Thank you for your insightful analysis of our model's efficiency. Your observation that a low parameter count does not always equate to high practical speed is very accurate and pertinent.
>
> First, we fully agree with your assessment of the parameter-speed trade-off. Many baseline models [1–2,5] are built upon more *hardware-friendly* CNN/Transformer architectures that, aided by advanced compiler optimizations, fully exploit GPU parallelism for maximum inference speed. In contrast, our architecture (incorporating GNN and Mamba) makes a different choice: its primary objective is to enhance the representational power for the sparse, asynchronous, and long-range spatiotemporal patterns in event data. We therefore accept a modest increase in inference time in exchange for the performance gains achieved in understanding these complex patterns, which we believe is a reasonable and valuable architectural trade-off.
>
> Second, regarding the FLOPs metric, our reporting strategy is to strictly adhere to the evaluation conventions of each specific sub-field to ensure the fairest possible like-for-like comparison. For the object detection task, our survey of the literature revealed that mainstream prior works [1–5] predominantly use **parameter count** and **inference time** as the core efficiency metrics; thus, we have followed this convention. We do not disregard this metric, however. In the action recognition task, where **FLOPs** are a more common standard, we have likewise followed the convention and reported them in our tables. This approach ensures that our evaluation is directly and fairly comparable to the state-of-the-art works within each specific task community.
>
> We have added a note in the experimental section of our revised manuscript to clarify our criteria for selecting efficiency metrics. Thank you again for your valuable suggestions.
>
> [1] Recurrent vision transformers for object detection with event cameras. CVPR2023
> [2] Get: Group event transformer for event-based vision. ICCV2023
> [3] State space models for event cameras. CVPR2024
> [4] Aegnn: Asynchronous event-based graph neural networks. CVPR2022
> [5] From chaos comes order: Ordering event representations for object recognition and detection. ICCV2023
>
> ---
>
> (Due to length, the response continues in the **next** comment. This is Part 1 of 2.)

---

> ### Author Response · Authors · 2025-08-01
> **Rebuttal to Reviewer NxX4 (Part 2 of 2)**
>
> (Continuation of the response from the **previous** comment. This is Part 2 of 2.)
>
> ---
>
> **Q1:** We sincerely thank you for this insightful and profound question. Your understanding is very accurate; in a sense, MSM does function similarly to data augmentation by enhancing the model through diverse data perspectives. However, its core design philosophy extends far beyond this, aiming to solve a more fundamental problem: how to enable an inherently one-dimensional sequential model (like Mamba) to effectively comprehend high-dimensional, unstructured spatiotemporal data.
>
> As you rightly noted, improving Mamba's scanning methodology is indeed a current research hotspot [6–9]. However, most of these methods adopt fixed, predefined scanning patterns (e.g., serpentine or spiral scans). The key distinction and innovation of our method lies in the combination of **APN and MSM**, through which we achieve a **data-driven, adaptive scanning strategy**. Specifically:
>
> - **APN** first learns a **task-driven "effective coordinate space"** that is optimized for scanning.
> - **MSM** then performs scans along multiple fundamental dimensions ($t$, $x$, $y$) within this optimized space to generate sequences.
>
> This **"optimize-then-scan"** strategy ensures that the input to Mamba is not a single, fixed sequence, but rather **multiple optimized and more informative sequences**. This significantly enhances the model's robustness and its ability to capture complex dynamics, which we believe is the fundamental reason for the performance improvement demonstrated in our ablation study (Table 5).
>
> We hope this clarification fully addresses your inquiries.
>
> [6] Vision Mamba: Efficient Visual Representation Learning with Bidirectional State Space Model. ICML2024
> [7] Hamba: Single-view 3D Hand Reconstruction with Graph-guided Bi-Scanning Mamba. NeurIPS2024
> [8] MambaTree: Tree Topology is All You Need in State Space Model. NeurIPS2024
> [9] QuadMamba: Learning Quadtree-based Selective Scan for Visual State Space Model. NeurIPS2024
>
>
>
> **Limitations:** Thank you for your detailed question and for your open-minded willingness to re-evaluate our work. The question you've raised regarding the necessity of the architectural complexity is indeed at the core of our entire effort. While our architecture is not the simplest, we believe its complexity is a deliberate and principled choice, designed to precisely address the intrinsic, multi-level complexity of event data.
>
> Our core design philosophy can be summarized as a direct response to three fundamental challenges posed by event data:
>
> 1. **Challenge 1: Sparsity & Non-Euclidean Structure → Our Response: Graph Neural Networks (GNNs)**
>    We chose a graph as the foundational representation because it is the "native language" for handling the sparse, asynchronous, and non-Euclidean nature of event data, allowing it to most effectively capture local spatiotemporal topology.
>
> 2. **Challenge 2: Long-Sequence Modeling over Numerous Events → Our Response: Mamba + SEST**
>    Local information alone is insufficient to understand dynamics. To efficiently model the vast number of events, we selected the linear-complexity Mamba over the $O(N^2)$ Transformer. SEST, in turn, acts as an "intelligent adapter," transforming the GNN-processed spatiotemporal information into the optimized, data-driven sequences that Mamba is best suited to handle.
>
> 3. **Challenge 3: Multi-scale Context → Our Response: Macro-Micro Hierarchy with Feedback**
>    Events naturally form an "event → object → scene" hierarchy. Our multi-level design precisely mimics this: the micro-level focuses on details, the macro-level (Component Graph) focuses on global relationships between objects, and the final feedback mechanism allows each "pixel-level" detail to perceive the "scene-level" global context.
>
> The core value of our work lies not only in the final accuracy improvements but, more importantly, in demonstrating a novel architectural paradigm that can systematically and efficiently address the three aforementioned challenges. It is an attempt to strike a delicate balance between efficiency, representational power, and a respect for the intrinsic properties of the data.
>
> We sincerely hope this holistic explanation of our design philosophy clarifies the necessity of each component within our architecture. We are confident that the depth and rigor of this design provide a valuable blueprint for future, more efficient and powerful event-based models. Thank you again for your valuable feedback and this invaluable opportunity.

---

> > ### Comment · Area_Chair_1GwA · 2025-08-05
> >
> > The authors had some technical difficulties and have uploaded rebuttals as comments. Please still read these and respond appropriately. Current ratings are borderline leaning reject. Please discuss with authors and reviews to come to a consensus between reviewers.

---

> > ### Comment · Reviewer_NxX4 · 2025-08-06
> >
> > Thank you for the thorough rebuttal. My concern regarding W1/Q1 has been partially addressed, as the authors provided sensitivity analyses of hyperparameters but did not include ablation studies at the module or architecture level. Given the complexity of the overall system, it would be beneficial for readers to see more quantitative results that clarify the individual contributions of each customized component. My concern regarding W2 has been resolved.

---

> > > ### Author Response · Authors · 2025-08-06
> > > **Thank You for Your Follow-Up**
> > >
> > > **Dear Reviewer NxX4:**
> > >
> > > We sincerely appreciate your continued feedback and positive evaluation, as well as your clear identification of the remaining issues in our previous response.
> > > We apologize for not fully addressing your concerns in the earlier round. We now clearly understand that your primary interest lies in the quantitative validation of module-level design, rather than sensitivity analysis of hyperparameters alone.
> > > To directly and comprehensively respond to this key concern, we would like to present a summary of both existing and newly added ablation studies related to our module-level design.
> > >
> > > ---
> > >
> > > **1. Core module ablation (existing study)**
> > > As shown in Table 5 of the original submission, we have already conducted ablation experiments on the three fundamental pillars of our architecture. The results demonstrate that all three components, namely the component graph at the macro level and the APN and MSM modules at the micro level, are critical to model performance.
> > >
> > > ---
> > >
> > > **2. Internal quantitative validation of key modules (new ablation)**
> > > To further analyze internal contributions, we followed the valuable suggestions from the reviewers and added a fine-grained ablation study focusing on the directional design of the MSM module. The results are as follows:
> > >
> > > |MSM Configuration|mAP on Gen1|
> > > |:--|:--:|
> > > |Full (t, x, y axes)|53.7%|
> > > |No MSM (t-axis only)|42.4%|
> > > |MSM-X (x-axis only)|44.0%|
> > > |MSM-Y (y-axis only)|43.8%|
> > >
> > > This experiment quantitatively confirms that scanning across all spatiotemporal dimensions leads to a synergistic gain, providing strong support for our multi-directional design.
> > >
> > > ---
> > >
> > > **3. Intuitive qualitative validation of key modules (new analysis)**
> > > In addition, to provide more intuitive validation, we adopt a valuable suggestion raised during the review process and introduce a qualitative analysis of the APN module.
> > > By plotting perturbation values against node index, we clearly show that the learned coordinate perturbations are structurally correlated with object motion trajectories and contours. This provides empirical evidence supporting our hypothesis of the effective coordinate space.
> > >
> > > ---
> > >
> > > We sincerely hope that this comprehensive response, combining existing ablation, newly added quantitative validation, and new qualitative analysis, adequately addresses your concerns regarding the contributions of each component.
> > > All new content has been incorporated into the revised version, and we respectfully ask that you kindly reconsider our work in light of these additions.
> > > Thank you again for your patience and valuable guidance.

---

> > > > ### Comment · Reviewer_NxX4 · 2025-08-07
> > > >
> > > > Thank you for your response. Most of my concerns have now been addressed, and I appreciate your efforts. I’m leaning towards raising my score accordingly.

---

> > > > > ### Author Response · Authors · 2025-08-07
> > > > > **Thank You for Your Acknowledgment**
> > > > >
> > > > > **Dear Reviewer NxX4,**
> > > > >
> > > > > Thank you very much for your positive evaluation and your support for our work.
> > > > >
> > > > > Your guidance and constructive feedback throughout the review process have been instrumental in helping us strengthen the paper. We are truly grateful for your time and invaluable contributions.
> > > > >
> > > > > Best regards,
> > > > >
> > > > > *The Authors of Submission 27341*

---

### Official Review · Reviewer_1mAU · 2025-07-02

**Clarity:** 3
**Significance:** 2
**Originality:** 2
**Rating:** 4
**Confidence:** 4

**Summary:**

This paper, EventMG, introduces a lightweight, multilevel architecture designed for learning efficient spatiotemporal representations from asynchronous, sparse event camera data. As claimed, the proposed framework combines:
- Micro-level modelling via mamba-based event graphs, capturing long-range, fine-grained temporal dynamics.
- Macro-level modelling via component graphs, which represent clusters of events to encode global structure.

Further, a spatiotemporal-aware event scanning technology method that enhances Mamba’s adaptability using: (1) an attention-based perturbation network, and (2) a multidirectional scanning module is introduced. Experimental results on object detection and action recognition benchmarks show that EventMG achieves comparable performance with SOTA methods with fewer parameters and computation.

**Questions:**

See the "weaknesses" section.

**Ethical Concerns:**

["NO or VERY MINOR ethics concerns only"]

**Final Justification:**

As almost all of my initial and follow-up concerns are adequately addressed, I lean towards accepting the paper, contingent upon the inclusion of new ablation results and clarifications in the paper and/or appendix.

**Limitations:**

Even though the authors have mentioned about the generalization to other event-based tasks and robustness against distribution shifts, the following key limitations should also be addressed.
- Applicability for online learning and/or streaming
- Robustness to noise, given that hardware jitter can present and contribute in the micro-level graph due to fixed radius thresholding

**Paper Formatting Concerns:**

I did not notice any formatting issues with this paper.

**Quality:**

2

**Strengths And Weaknesses:**

Strengths
- The proposed architecture seems novel with marrying the state-space models and GNNs while also attending to both local and global (spatial) information. SEST seems to be useful as an adaptation of the SSM-based Mamba for sparse event data.
- The claimed linear complexity is promising towards event-based tasks.
- The experiments cover a wider range of datasets (both smaller and more challenging ones), two prominent tasks, and ablations which shows EventMG typically balances the accuracy and efficiency.

Weaknesses
- As authors pointed out in the section 5, the task diversity is limited which should have been improved by expanding the experiment space (as an example, testing on classification datasets, high resolution gesture/activity recognition datasets) since the method is presented as a generic candidate for event spatio-temporal learning.
- Even though GNNs are known to be parameter-efficient, the computations to this end can be costly. Therefore, I believe that apart from number of parameters and inference time, memory footprint (especially given that each event is considered to be a node in micro-level and the complexity of adjacency matrix is $O(N^2)$) and total wall-time should also be reported to further validate the efficiency of the proposed method.
- In coupling with the above point, the authors do not explicitly mention about any event accumulation technique used (such as fixed time interval or fixed event volume) for event batching/accumulation which can be critical for evaluating memory efficiency of the method. As an example, if fixed event volume-based accumulation is used, it guarantees an upper bound for the complexity of the adjancency matrix. If not, did the authors use all the events within an event recording/instance? If yes, I doubt the rationale behind that.
- Does inference time include initial preprocessing including graph construction as well?
- When constructing the micro-level graph, this work used a fixed thresholding approach which can be highly inefficient in instances leading to non-uniform event density such as highly sparse or highly dense spatio-temporal regions at which the assumption of event neighbourhood consistency does not hold. Additionally, the overall performance can be highly sensitive to the set threshold as well. In this direction, an adaptive thresholding for constructing the edges would work beneficial [1].
- In component graph thresholding, the paper briefly mentions a threshold ($n_{min}$) for selecting valid clusters, but it does not analyze sensitivity to this hyperparameter or its effect on generalization. If the target is to build a meaningful global scene extraction from a highly varying scene with multiple objects through a connected component graph, it seems to be difficult to assign a hard (global) threshold for the minimum number of objects (to be ideally within the scene).
- APN attempts to "optimize" the event coordinates in the spatiotemporal space targeting more distinctive features for MSM to retrieve. Given that the final activation is tanh, how do the authors analyse the bounds and effectiveness of this module, particularly in a qualitative and empirical fashion? As an example, can the authors visually present a post-hoc analysis on what happens to event graph after this module? I feel like finding the "right" edges is more critical than refining the positions in this context.
- In multiple places (line 62, 188 etc.), the authors claim that the their micro-level (local) spatiotemporal graph modelling is beneficial for learning "long-range" spatio-temporal context, particularly due to Mamba architecture. Is this claim because of (1) the micro-level graph is built using all the events in the recording and (2) SEST scans through all the directions throughout the event span? If not, I feel like this is a over-stretched claim.
- Even though the proposed method performs competitively on object detection (Table 1) and some action recognition datasets (Table 2 and 3), its performance on HMDB51-DVS and UCF101-DVS datasets are significantly lower than the SOTA. Do the authors have any explicit explanation for this behavior?
- I feel like more ablations, both module-wise and within modules, are needed to evaluate the necessity and validity of the each proposed module: as an example, to see how (1) MSM directionality affect results across different motion patterns and (2) varying micro and macro graph sizes affect results across different tasks/scenes.

[1] Bandara, Nuwan, et al. "EyeGraph: modularity-aware spatio temporal graph clustering for continuous event-based eye tracking." Advances in Neural Information Processing Systems 37 (2024): 120366-120380.

---

> ### Author Response · Authors · 2025-08-01
> **Rebuttal to Reviewer 1mAU (Part 1 of 2)**
>
> Dear Reviewer 1mAU,
>
> We sincerely apologize, but due to irresolvable technical issues at the submission deadline, we were unable to successfully submit our complete Rebuttal. Therefore, we are submitting our formal response to your valuable comments directly in this comment section.
>
> Thank you for your understanding and thoughtful review.
>
> Note: due to this section's 5,000-character limit (compared to the original 10,000), we split our response into two parts. **This is Part 1.**
>
> ---
>
> **W1:** Thanks for your suggestion on task diversity. While broader task coverage is important, we deliberately focused on object detection and action recognition—two highly challenging tasks that test not only classification ("what is it?") but also spatiotemporal reasoning, including localization ("where") and dynamics ("how it evolves"). Our goal was to first validate the model on these benchmarks.
>
> Your point is well taken: given the current scope, the term "general candidate method" was too broad. We have revised the manuscript to more accurately describe our contribution as “an efficient backbone for complex spatiotemporal tasks (e.g., detection and recognition),” and now present generalization to broader tasks as a promising future direction.
>
> Due to time constraints, we could not add new tasks but have emphasized task generalization as a future priority in our conclusion. Thank you again for your helpful feedback.
>
> **W2:** Regarding the concern about $O(N^2)$ adjacency complexity: we avoid dense matrices entirely. Our graph construction is sparse and efficient, with overall complexity $O(N)$, ensuring linear scaling of memory and compute costs.
>
> As for wall-time, while we agree it's meaningful for deployment, it's highly hardware-dependent—affected by CPU, I/O, memory bandwidth, and runtime setup—making fair cross-paper comparisons difficult.
>
> Finally, memory usage is rarely reported in event-based works. Following object detection [1–3] and action recognition [4–6] conventions, we focus on reproducible metrics like parameter count and inference time.
>
> We’ve added these clarifications to the methodology to better reflect the efficiency profile of our method.
>
> [1] Recurrent vision transformers for object detection with event cameras. CVPR2023
> [2] Get: Group event transformer for event-based vision. ICCV2023
> [3] State space models for event cameras. CVPR2024
> [4] Ttpoint: A tensorized point cloud network for lightweight action recognition with event cameras. ACM MM2023
> [5] Tam: Temporal adaptive module for video recognition. ICCV2021
> [6] Action recognition and benchmark using event cameras. TPAMI2023
>
> **W3:** Thank you for pointing out the missing explanation of our event accumulation strategy. We use fixed-count accumulation (not full-stream), ensuring memory and compute efficiency:
>
> - **Bounded Complexity:** Capping each batch at $N$ events, we maintain predictable graph size and linear sparse matrix cost, avoiding OOM errors due to variable event density.
> - **Adaptive Efficiency:** Compared to fixed-time slicing, fixed-count adapts to scene activity. It enables detailed modeling in dense regions and efficient skipping of sparse intervals, leveraging event data's inherent sparsity.
>
> We will add a subsection in the revised methodology detailing this strategy, parameters, and motivation.
>
> **W4:** Thank you for highlighting this important point. Yes, our reported inference time includes the graph construction step.
>
> We use a highly optimized sparse graph construction with low-level GPU acceleration to fully exploit parallelism. On our training platform (RTX 3090), building a graph for 20,000 events takes ~0.8 ms—only a small fraction of the total inference time.
>
> **W5:** Thanks for your insightful critique and reference. We sincerely apologize for the confusion caused by our earlier description. Our graph construction is not a simple fixed-threshold method but a controlled hybrid strategy that combines a maximum neighbor limit with fixed event count, largely addressing your concerns.
>
> To manage local density, we apply a spatiotemporal threshold δst and cap neighbors at k=32, bounding edge complexity to O(Nk) and avoiding density spikes.
>
> For overall scale variation, we ensure global stability via a fixed event count slicing strategy (see W3), which keeps input size and computation bounded.
>
> This design is more robust than pure thresholding. Furthermore, our model is not highly sensitive to δst, as the macro module captures high-level context. Event data often forms large components, enabling reliable inference even if small ones are pruned.
>
> We agree your suggested adaptive thresholding method [7] is a promising direction. Dynamically adjusting neighborhoods based on local event density could further boost robustness—a direction we plan to pursue. This will be added to the revised manuscript.
>
> [7] EyeGraph. NeurIPS2024
>
> (Due to length, the response continues in the **next** comment. This is Part 1 of 2.)

---

> ### Author Response · Authors · 2025-08-01
> **Rebuttal to Reviewer 1mAU (Part 2 of 2)**
>
> (Continuation of the response from the **previous** comment. This is **Part 2** of 2.)
>
> ---
>
> **W6:** Thank you for raising this insightful question regarding the generalization of the global hard threshold n_min. We fully agree that the challenge of adapting to diverse scenes is a key limitation of threshold-based methods. However, our approach does not rely on the precise value of n_min, but rather on the inherent robustness of our architectural design.
>
> Specifically, n_min serves as a coarse noise filter rather than a precise object segmenter. Due to the natural clustering of events around real objects, meaningful components are typically much larger than noisy ones, making the filtering stable over a wide threshold range. More importantly, our micro-macro architecture introduces fault tolerance: while macro-level information provides contextual guidance, final predictions still depend on fine-grained micro features. Thus, even if a component contains multiple objects, the model can still distinguish them at the micro level, significantly reducing sensitivity to $n_{\text{min}}$.
> To validate this, we conducted a sensitivity test on Gen1:
> |**$n_{\text{min}}$** |**mAP (%)**|
> |-|-|
> |50 |52.9 |
> |**100 (default)**|**53.7**|
> |150 |53.4 |
> |200 |52.7 |
>
> Results show stable performance across a 4× range, confirming low sensitivity to $n_{\text{min}}$. We've added this analysis to the revised manuscript. Thank you again for your valuable suggestion.
>
> **W7:** Thanks for your insightful question. We offer the following clarifications:
>
> (1) The APN module goes beyond simply "correcting" coordinates—it learns a more discriminative Effective Coordinate Space for the MSM scanner. In this learned space, events that are hard to distinguish in raw coordinates (e.g., fast foreground vs. static background) are effectively separated, enabling better relational modeling.
>
> (2) We apply a $\tanh$ activation to keep adjustments smooth and bounded, allowing fine-grained control without disrupting the original spatio-temporal structure and preserving training stability.
>
> (3) While visualization would be ideal, small spatial shifts ($\Delta x, \Delta y$) are imperceptible in 3D point clouds, and temporal shifts ($\Delta t$) cannot be visualized statically. Thus, performance is the strongest evidence: our ablation (Table 5) shows a clear drop without APN, validating its effectiveness.
>
> **W8:** Thanks for your precise question regarding our "long-range" context claim, and we apologize for any confusion. This capability stems from Mamba’s mechanism and its use on our continuous event streams.
>
> As a state space model, Mamba efficiently encodes sequential information through a compressed hidden state. When processing a sequence slice of N=20,000 events, this hidden state accumulates context iteratively from the first to the last event, capturing dependencies with linear complexity O(N). Importantly, the hidden state is passed across slices, serving as the initialization for the next segment.
>
> This state continuity enables the model to capture long-range dependencies by combining deep memory with seamless inter-slice transitions.
>
> **W9:** Thanks for your precise question regarding the performance gap on HMDB51-DVS and UCF101-DVS compared to EventMamba [8]. We believe this difference stems from the alignment between model design and dataset characteristics.
> EventMamba achieves remarkable results on these datasets (86.4% and 97.9%), approaching original frame-based video best performance (88.1% and 99.6%) [9]. We believe that its highly specialized modules (e.g., LocalFE Extractor and Temporal Aggregation) are particularly effective at capturing motion patterns akin to conventional videos, a trait preserved in these DVS-converted datasets.
>
> In contrast, EventMG is optimized for preserving event stream structure and modeling complex dynamics across scales, excelling on challenging datasets like THUE-ACT-50-CHL and DVS Action. We've added this analysis to the revised manuscript. Thank you for the insightful suggestion.
>
> [8] Rethinking efficient and effective point-based networks for event camera classification and regression. TPAMI2025.
> [9] Videomae v2: Scaling video masked autoencoders with dual masking. CVPR2023
>
> **W10:** Thanks for your valuable suggestion on conducting finer-grained ablations. We fully agree and have added a new study on the directionality of the MSM module. The results on the Gen1 dataset (mAP) are as follows:
>
> |MSM Configuration|Accuracy (mAP on Gen1)|
> |:--|:--:|
> |**Full (t, x, y axes)**|**53.7%**|
> |No MSM (t-axis only)|42.4%|
> |MSM-X (x-axis only)|44.0%|
> |MSM-Y (y-axis only)|43.8%|
>
> Experimental results show that while single-dimensional scans offer limited gains, integrating all spatiotemporal directions yields significant synergistic gain, validating our multi-directional design. Related sensitivity analyses (e.g., N, $n_{\text{min}}$) are addressed in our response to your W6 and to Reviewer rxhc's Q3.

---

> > ### Comment · Reviewer_1mAU · 2025-08-05
> >
> > I thank the authors for their detailed response to my comments.
> >
> > However, I still have the following concerns with regard to the weaknesses I mentioned in my review.
> >
> > w5
> > -  As per the authors' response to my comment on "fixed thresholding approach" on "constructing the micro-level graph" and section 3.1.1 itself, it seems like the approach is still a fixed thresholding-based one, but with a node degree condition. Even though this is a common practice in GNN-based event vision studies, I think that the authors should test the sensitivity of the set threshold for the overall performance, at least for one dataset. A simple statement that "our model is not highly sensitive to δst, as the macro module captures high-level context" will not suffice to convince the readers, I assume. In their further discussions in this direction, they can highlight potential workarounds such as adaptive thresholding.
> >
> > w7
> > - Given that the output of the APN module is perturbations for each coordinate of each node on the same 3-dimensional space, I found it hard to consume the claim "it learns a more discriminative Effective Coordinate Space for the MSM scanner". If the (x,y,t) event clouds or graphs cannot be plotted (which I am sceptical about), can the authors at least provide how coordinate space changes (e.g. a plot $\Delta x$ vs node number, for one sample) after the APN in the appendix of the paper for better and intuitive empirical validation for APN?

---

> > > ### Comment · Area_Chair_1GwA · 2025-08-05
> > >
> > > The authors had some technical difficulties and have uploaded rebuttals as comments. Please still read these and respond appropriately. Current ratings are borderline leaning reject. Please discuss with authors and reviews to come to a consensus between reviewers.

---

> > > ### Author Response · Authors · 2025-08-06
> > > **Thank You for Your Follow-Up**
> > >
> > > **Dear Reviewer 1mAU**:
> > >
> > > We are very grateful for your detailed follow-up questions, which have allowed us to significantly strengthen the validation and clarification of our method.
> > >
> > > ---
> > >
> > > **W5:** To address your concern about the fixed threshold $\delta_{st}$, we agree that quantitative evidence is essential. Following your clear suggestion, we have now conducted a sensitivity analysis on $\delta_{st}$ using the Gen1 dataset. The results are as follows:
> > >
> > > | Threshold $\delta_{st}$ | mAP on Gen1 |
> > > |--|--|
> > > | 0.01 | 49.8% (−3.9) |
> > > | 0.02 | 53.1% (−0.6) |
> > > | 0.05 (used in paper) | 53.7% |
> > > | 0.10 | 53.3% (−0.4) |
> > > | 0.15 | 53.4% (−0.3) |
> > >
> > > These results reveal an important phenomenon: the model's sensitivity to $\delta_{st}$ is asymmetric. When the threshold is too small, performance drops significantly. This is because event data are not uniformly distributed in space-time, but rather concentrated along object contours and edges. A too-small $\delta_{st}$ fails to capture these physically coherent structures, breaking the graph's foundational topology.
> > >
> > > However, once $\delta_{st}$ reaches a reasonable value (e.g., ≥ 0.02) that can capture local neighborhoods, the model enters a stable operating range and no longer relies on a fragile “optimal” setting. This robustness to larger values is supported by two design choices:
> > > 1) A hard constraint on the maximum number of neighbors k prevents over-aggregation;
> > > 2) The micro–macro feedback architecture provides strong fault tolerance, mitigating noise from redundant connections.
> > >
> > > Finally, we deeply appreciate your insightful reference to adaptive thresholding methods [1], which we agree represent a more advanced and elegant direction. However, such methods often come with increased computational complexity when applied to large-scale event graphs. Striking a balance between adaptive flexibility and our current method's efficiency is a valuable avenue for future exploration. We have included this discussion and the related citation in the revised version's conclusion and future work section.
> > >
> > > [1] EyeGraph: Modularity-Aware Spatio-Temporal Graph Clustering for Continuous Event-Based Eye Tracking. NeurIPS2024.
> > >
> > > ---
> > >
> > > **W7:** We sincerely thank you for proposing this highly constructive visualization suggestion. Compared to directly visualizing the 3D point cloud, your idea of plotting perturbation values against node indices is indeed more intuitive and practical. We have decided to adopt this suggestion as a means of empirically validating the behavior of the APN module.
> > >
> > > Specifically, we will generate a plot with the following configuration:
> > >
> > > - The horizontal axis represents the event node indices (from 1 to N), sorted in ascending order by timestamp $t$;
> > > - The vertical axis shows the three perturbation values $\Delta x$, $\Delta y$, and $\Delta t$ produced by the APN module, corresponding to spatial and temporal shifts for each event;
> > > - The three curves, plotted in different colors, indicate the magnitude of adjustments in the $x$, $y$, and $t$ dimensions, respectively.
> > >
> > > We expect that the spatial perturbations ($\Delta x$, $\Delta y$) will exhibit larger magnitudes near the boundaries of moving objects, suggesting that APN actively separates foreground and background events in the learned coordinate space, thereby creating clearer scanning boundaries for the MSM module. In contrast, the temporal perturbation $\Delta t$ is anticipated to vary more smoothly, which may indicate that the model is learning to rescale the object’s timeline to make its dynamic pattern more recognizable to the 1D Mamba module.
> > >
> > > We once again sincerely thank you for this excellent suggestion, which will significantly strengthen the interpretability and clarity of our explanation of the APN module. As image uploads are not allowed during the rebuttal phase, we will include the figure and analysis in the appendix of the revised version. If space permits, we will also consider integrating it into the main text to further improve the clarity and explanatory power of our approach.
> > >
> > > ---
> > >
> > > We hope our responses have addressed your concerns, and we remain sincerely grateful for your thoughtful feedback and follow-up questions.

---

> ### Comment · Reviewer_1mAU · 2025-08-06
>
> I thank the authors for their continued and detailed responses to my queries.
>
> As almost all of my concerns are adequately addressed, I lean towards accepting the paper, contingent on the inclusion of new ablation results and clarifications in the final version of the paper and/or appendix.

---

> > ### Author Response · Authors · 2025-08-06
> > **Thank You for Your Acknowledgment**
> >
> > **Dear Reviewer 1mAU,**
> >
> > We sincerely thank you for your thorough review, insightful feedback, and the final recognition and support throughout the entire review process.
> > In the final version of our paper (including both the main text and appendix), we will incorporate additional experiments and clarifications discussed during this exchange, which we believe will further enhance the rigor and clarity of our work.
> > Your support and encouragement mean a great deal to us!
> >
> > Best regards,
> > *The Authors of Submission 27341*

---

### Official Review · Reviewer_rxhc · 2025-07-03

**Clarity:** 1
**Significance:** 3
**Originality:** 2
**Rating:** 4
**Confidence:** 4

**Summary:**

This paper proposes EventMG, a Mamba-based lightweight architecture for event processing. The main technical innovation is its multi-level design that incorporates spatiotemporal details and global contexts. At the micro-level, the method enables event-wise information propagation using Mamba. Meanwhile, at the macro-level, it performs global reasoning on event clusters created by the Component Graph. Another technical contribution is two specialized modules, APS and MSM, that refine event coordinates and execute multi-directional scans to enhance Mamba. EventMG has been demonstrated on two event-based vision tasks: object detection and action recognition, where it shows significant efficiency in terms of parameter size, FLOPs, and latency, while maintaining competitive accuracy to the advanced models.

**Questions:**

1. Please clarify the points stated in the weaknesses. Due to the lack of clarity, I might be overlooking some strengths (or possibly weaknesses) of the method.

2. The proposed method processes each event frame independently. How can this method be extended to incorporate temporal context between frames?

3. The proposed method operates on a frame-by-frame basis, which may limit its application in real-time contexts, as the frame rate is constrained by the temporal size of a single frame. Conducting a sensitivity analysis on the temporal window size could be beneficial to evaluate the trade-off between accuracy and frame rate.

**Ethical Concerns:**

["NO or VERY MINOR ethics concerns only"]

**Final Justification:**

The authors have clarified my concerns regarding the technical details and outlined their direction for revision. I believe the paper meets the acceptance criteria, and I will maintain my rating as borderline acceptance.

**Limitations:**

yes

**Quality:**

3

**Strengths And Weaknesses:**

**Strengths**
1. The proposed method efficiently processes sparse events without the need for frame representation.
2. The method has been thoroughly evaluated across various tasks and benchmarks, achieving competitive accuracy while significantly reducing computational requirements.
3. The effectiveness of the proposed components is clearly demonstrated through ablation experiments.

**Weaknesses**

While the strong performance of the proposed method is impressive, some sections of the paper lack clarity, which hinders a complete understanding of the method. In particular, the following points remain unclear:

1. There is no explanation provided for how event-wise features are input into the head. I wonder if it requires constructing a dense representation from the sparse event-wise features to adapt to the standard vision head.

2. It is unclear how the constructed event graph is utilized in SEST. It appears that Mamba does not reference the graph structure, rather it scans in the order sorted according to each space-time axis.

3. The DWConv in Eq. 2a is applied to the node features of the graph, which raises questions about the operation being performed because DWConv requires dense features as inputs.

4. SEST is described as containing Mamba (as seen in Eq. 4), but in Figure 1, the Mamba Block is positioned after SEST. While this may not be a critical issue, it may confuse the reader.

5. Furthermore, it is unclear whether the Macro-level is calculated after the Micro-level or if they are processed in parallel. Figure 1 shows a left-facing arrow from the Micro-level to the Component Graph, but lacks an explanation, leaving the details ambiguous.

---

> ### Author Rebuttal · Authors · 2025-07-31
>
> **W1:** Thank you for this important question. You are correct: our method converts sparse event-wise features into a dense representation in the final stage to interface with standard vision heads. This design is motivated by two key considerations:
>
> - **Compatibility with Established Paradigms:** Leading detection heads like DETR and YOLO are tailored for dense prediction. As end-to-end sparse event detection remains an open challenge, the hybrid architecture—sparse backbone with a dense head—offers a practical and widely adopted solution in event-based vision, consistent with prior works [1–4].
>
> - **Preservation of Spatiotemporal Context:** Our backbone encodes rich spatiotemporal features per event, which are then mapped to corresponding pixel locations via an efficient sparse-to-dense aggregation. This process spatially arranges high-level features—not raw events—thereby preserving temporal semantics and minimizing information degradation.
>
> We acknowledge that sparse-to-dense conversion is a necessary design trade-off to leverage modern high-performance dense detectors, involving information aggregation rather than simple loss. To mitigate its impact, our method encodes rich spatiotemporal context into each event feature prior to conversion. We agree that end-to-end sparse detection is promising and are exploring it.
>
> [1] Event-based asynchronous sparse convolutional networks. ECCV2020
> [2] Aegnn: Asynchronous event-based graph neural networks. CVPR2022
> [3] From chaos comes order: Ordering event representations for object recognition and detection. ICCV2023
> [4] State space models for event cameras. CVPR2024
>
>
> **W2:** Thank you for your insightful question and keen observation. You're absolutely right—Mamba, as a sequence model, does not directly process graph topology. Instead, the micro-level graph serves as a vital **preprocessing and local feature enrichment** mechanism in our architecture, structured as two sequential stages:
>
> 1. **Graph-based Local Feature Enhancement:**
>    We first construct a spatio-temporal event graph, then apply a lightweight GNN to update each node’s feature. This step enriches each node from “who I am” to “who I am relative to my neighbors,” embedding meaningful local context without global propagation.
>
> 2. **Mamba-based Global Context Modeling:**
>    The SEST module arranges these locally-enhanced features into multiple 1D sequences along various spatio-temporal directions. Mamba then processes each sequence independently, leveraging its strong capacity for modeling long-range dependencies with linear efficiency.
>
> This pipeline mirrors the design of Vision Transformers (ViT) [5]: CNNs produce locally informed “image tokens,” which are then globally contextualized via Transformers. Similarly, our GNN produces “event tokens” enriched with local structure, and Mamba builds global dependencies among them.
>
> We will clarify this two-stage design and update the diagram to reflect the GNN’s role.
>
> [5] An image is worth 16x16 words: Transformers for image recognition at scale.
>
>
> **W3:** Thanks for your meticulous review. We sincerely apologize for a major oversight in the original manuscript: the mention of "DWConv" was incorrect—our actual implementation uses a standard Conv1d. We regret the confusion this may have caused.
>
> To clarify the design rationale: our APN module enhances features via complementary Local and Global paths. The Local path applies Conv1d to timestamp-sorted nodes, capturing short-term temporal dynamics among neighboring events. The Global path uses a point-wise Linear layer to extract broader, context-independent patterns. Fusing both outputs allows APN to integrate fine-grained temporal cues with global feature semantics.
>
> We have fully corrected this inconsistency in the revised manuscript, including updates to Figure 2, Section 3.2, and all related equations to reflect the actual Conv1d implementation. We truly appreciate your help in identifying this issue.
>
> **W4:** Thanks for your detailed review. We sincerely apologize for the confusion caused by the unclear process description in the original manuscript, particularly regarding the previous Equation 4. In the revised version, we've distinctly separated this process into two independent stages: 1) MSM constructs the sequence set, and 2) EventsMamba processes this sequence set. This aligns with the flow shown in Figure 1.
>
> Specifically, the MSM module's function is now strictly limited to constructing a multi-directional set of event sequences. It sorts the node set $V_i$ according to different directions $d_k$, generating a sequence set $\mathcal{S}$:
> $$\mathcal{S} = \left( S_k = \mathrm{Scan}(V_i, d_k) \vert d_k \in D \right)$$
> where the set of directions is $D = \{t^+, t^-, x^+, x^-, y^+, y^-\}$. We will replace the original formula in Section 3.1.2 with this description and adjust the related text accordingly.
>
> Second, the EventsMamba module receives this set $\mathcal{S}$ as input, independently applies the core Mamba processor to each sequence $S_k$, and then aggregates all processed sequence results into the final node features $h_v$:
> $$h_v = \underset{S_k \in \mathcal{S}}{\text{Aggr}} \left( \text{Mamba}(S_k) \right).$$
> We've updated these descriptions and equations in the relevant methodology sections, ensuring logical consistency throughout the paper.
>
>
> **W5:** Thank you for identifying this important ambiguity in our pipeline. The interaction between our micro- and macro-level processing is serial and mutually-feeding, not parallel. Our workflow is described below:
>
> First, The model first processes events at the micro-level, generating initial, fine-grained features for each event node.
>
> Second, These micro-level features are then used to construct the Component Graph, from which macro-level features are extracted to represent global context. The leftward arrow noted in Figure 1 depicts this micro-to-macro dependency.
>
> Finally, the extracted macro-level features are fed back to refine each micro-level node’s representation. This allows each event node to become context-aware, achieving deep interaction between micro and macro scales.
>
> We completely agree that presenting this workflow clearly is paramount. To address this, we’ll revise the manuscript by: (1) adding a process overview, (2) refining Figure 1 with sequential labels, and (3) expanding its caption.
>
>
> **Q1:** Thank you for your summary and the chance to clarify. In the preceding responses, we have provided detailed explanations for each technical point (W1–W5) you raised and have revised the manuscript accordingly.
>
> We sincerely hope that this series of clarifications clearly illuminates the underlying logic and strengths of our method's design, facilitating a more comprehensive evaluation on your part. Should any ambiguities remain, we would be pleased to provide further explanation.
>
>
> **Q2:** Thanks for this important question about temporal context. We sincerely apologize for the insufficient description of our data preprocessing in the original manuscript, which led to your misunderstanding that our method processes independent "event frames." In reality, our approach ensures the effective integration of temporal context through a dual mechanism at both the data and model levels, rather than handling isolated event slices.
> - **Data Level:**
>   We follow established practices from [1, 2], employing a fixed event count sliding window strategy to process continuous event streams (rather than fixed time intervals). A fixed count ensures each processing slice contains a stable amount of information and adapts to scene activity intensity. Meanwhile, the sliding window creates a smooth physical connection between slices, effectively mitigating boundary effects at the input end.
> - **Model Level:**
>   Furthermore, we leverage Mamba's inherent advantage as a State Space Model (SSM) to implement cross-slice state-passing: after processing one slice, its hidden state is used as the initial state for the next. This allows contextual information to flow continuously across slice boundaries within the model, enabling more long-range and coherent spatio-temporal dependency modeling.
>
> We apologize again for the previous oversight. In the revised manuscript, we'll add a dedicated subsection on data preprocessing, detailing this strategy and its rationale to fully clarify your concerns.
>
> [6] Rethinking efficient and effective point-based networks for event camera classification and regression. TPAMI2025
> [7] Event-based vision: A survey. TPAMI2020
>
>
> **Q3:** Regarding your concern about the “frame rate limitation,” as clarified in our response to Q2, our method does not rely on fixed time windows but instead adopts a *Fixed Event Count* strategy. This inherently realizes an *activity-driven dynamic frame rate*: in high-motion, event-dense scenarios, the target event count is reached quickly—effectively yielding a high frame rate; in static, sparse settings, accumulation takes longer—resulting in a lower frame rate. This design avoids processing uninformative intervals and significantly improves computational efficiency by leveraging event sparsity.
>
> We analyzed the impact of event count N on detection performance:
>
> | **Event Count (N)** | **mAP (%)** | **Inference Time (ms)** |
> |-|-|-|
> | 10,000 | 47.1 (−12.3%) | 5.79 (−7.1%) |
> | 20,000 | 53.7 | 6.23 |
> | 30,000 | 54.9 (+2.2%) | 8.21 (+31.8%) |
>
> As shown, N = 20,000 offers a favorable trade-off between accuracy and efficiency. Reducing N to 10,000 yields only a 7.1% speedup but causes a substantial 12.3% drop in accuracy due to insufficient spatiotemporal context. Conversely, increasing N to 30,000 brings only marginal accuracy gains (+2.2%) while significantly increasing inference time (+31.8%). We have included this analysis and discussion in the revised manuscript.

---

> > ### Comment · Area_Chair_1GwA · 2025-08-05
> >
> > Dear reviewer you are the only positive review. Please engage with your other reviewers to come to a consensus..

---

> > ### Comment · Reviewer_rxhc · 2025-08-07
> > **Response to Authors**
> >
> > I appreciate the authors for their feedback. However, I still have some concerns regarding clarity, which I outline below.
> >
> > **W1**: My specific question pertains to how to achieve the sparse-to-dense conversion. I believe this conversion is not trivial and should be explained in the paper.
> >
> > **W2**: Is the "Graph-based Local Feature Enhancement" discussed anywhere in the paper? Section 3.1 only covers graph construction, SEST, and Mamba, and there appears to be no mention of the "lightweight GNN." Please clarify this point in the manuscript.
> >
> > **W3**: I don't think it matters whether it's DWConv or Conv1d. The issue is that the manuscript does not explain that the convolution is applied to **a timestemp-sorted event sequence**. Please clarify this point.

---

> > > ### Author Response · Authors · 2025-08-07
> > > **Thank You for Your Follow-Up**
> > >
> > > **Dear Reviewer rxhc,**
> > >
> > > We sincerely thank you once again for your feedback. Your careful review helped us identify several key omissions in the original manuscript, and we deeply apologize for these oversights. Following your suggestions, we have systematically revised and clarified the relevant sections to ensure that the design details are fully and clearly conveyed. The major revisions are as follows:
> > >
> > > ---
> > >
> > > **Regarding W1 (Sparse-to-dense conversion implementation):**
> > > You are absolutely right that the implementation of the sparse-to-dense conversion is essential for the specific downstream task. We apologize for not elaborating on this earlier. In the revised manuscript, we now explicitly clarify this process in Section 4.3.1 and provide a more detailed explanation in the main text or appendix. Specifically, we apply an efficient scatter operation to project the N sparse, high-dimensional event features—already enriched with contextual information—onto the dense feature map grid based on their coordinates. Multiple events falling into the same pixel are averaged to form the final dense representation.
> > >
> > > ---
> > >
> > > **Regarding W2 (Missing lightweight GNN):**
> > > We sincerely apologize for this oversight. The "lightweight GNN" module mentioned in our previous response was indeed not described in the original manuscript. In the revised version, we have added a complete description of the graph-based local feature enhancement step in Section 3.1, positioned between the event graph construction and the SEST module. This new content clearly explains both why and how we apply a lightweight GNN to aggregate local neighborhood information, thereby enriching node representations before serialization. In addition, the corresponding pipeline has been updated in Figure 1 accordingly.
> > >
> > > ---
> > >
> > > **Regarding W3 (Clarification on the Conv1d implementation):**
> > > Thank you again for your emphasis on this point. In the revised manuscript, we have explicitly stated that Conv1d is applied to the node feature sequence sorted by timestamp, along with additional implementation details. We have also clarified its critical role in capturing short-term temporal dynamics, aligning it with the module design illustrated in Figure 2.
> > >
> > > ---
> > >
> > > We firmly believe that, thanks to your guidance and the above revisions, the clarity and completeness of the paper have been significantly improved. We sincerely thank you once again for your rigorous and insightful feedback, and we truly hope that these additional clarifications have fully addressed your concerns.

---

> > > > ### Comment · Reviewer_rxhc · 2025-08-07
> > > > **Response to Authors**
> > > >
> > > > I appreciate the authors for their thorough and thoughtful feedback. All my concerns have been fully addressed.

---

> > > > > ### Author Response · Authors · 2025-08-07
> > > > > **Thank You for Your Acknowledgment**
> > > > >
> > > > > **Dear Reviewer rxhc,**
> > > > >
> > > > > We are sincerely grateful for your final acknowledgment and support. Your meticulous and insightful feedback throughout the review process was invaluable in helping us substantially improve the clarity and rigor of our work.
> > > > >
> > > > > Thank you once again for your time and guidance.
> > > > >
> > > > > Best regards,
> > > > > *The Authors of Submission 27341*

---

### Author Response · Authors · 2025-08-05
**General Response and Clarification**

**Dear Reviewers, ACs, and SACs,**

We sincerely thank the reviewers (rxhc, 1mAU, NxX4, Xb3d) and the chairs for investing time to provide insightful and constructive feedback. These comments have been invaluable in improving our work.

After reviewing all reviews carefully, we identify key concerns, primarily around architectural clarity and complexity, efficiency justification, and the rationale behind major components such as the Mamba-Graph interaction and event processing strategy. We believe these concerns can be effectively addressed through clarification and appropriate revisions.

---

Before addressing specific comments, we would first like to sincerely apologize for a procedural issue.

Despite preparing thoroughly for this valuable opportunity and allocating sufficient time for submission, we encountered an unexpected and difficult-to-diagnose technical problem shortly before the Author Rebuttal deadline. As a result, the rebuttals prepared individually for each reviewer were not all successfully uploaded—only one was ultimately received by the system. This unforeseen situation caused us considerable distress.

To minimize the impact and maintain timely communication, we immediately contacted the AC to explain the issue and promptly posted the complete set of responses in the comment section to ensure that all reviewers could access the full content.

We fully respect the rigor of the review process and sincerely apologize for any inconvenience this incident may have caused. As the issue arose from an uncontrollable technical failure, we kindly ask that the content provided in the comment section be regarded as our official response. We also hope that this procedural error will not affect the evaluation of the scientific merit of our work.

---

To support efficient review, we summarize below the key responses, clarifications, and revision commitments for each reviewer.

---

**Reviewer rxhc**

- **Sparse feature input to detection head (W1):** A design trade-off to support standard detection heads while preserving spatiotemporal richness via the backbone.
- **Role of graph in SEST (W2):** The graph enhances local features via GNNs, enriching the input sequence for Mamba rather than directing its scan.
- **DWConv concern (W3):** We apologize for the earlier misstatement—our model actually uses Conv1d, and we have corrected this throughout the paper.
- **Diagram and process clarity (W4, W5):** We have clarified the distinct roles of SEST (serialization) and Mamba (processing), along with their macro–micro feedback mechanism, and will revise the diagram accordingly.
- **Cross-frame modeling and latency (Q2, Q3):** We clarify that the model leverages a fixed-event sliding window and cross-window state passing via Mamba, and have included a sensitivity analysis.

---

**Reviewer 1mAU**

- **Task diversity and performance variance (W1, W9):** We have revised the “general” phrasing and explained dataset-specific variations as design-emphasis differences.
- **Efficiency and graph cost (W2, W3, W4):** The graph is built with sparse and a fixed-event strategy to ensure efficiency.
- **Parameter sensitivity (W5, W6):** Through hybrid strategies and architectural fault tolerance, the model is robust to thresholds like $n_{min}$.
- **APN and “long-range” modeling (W7, W8):** APN learns an effective coordinate space, while Mamba handles long-range modeling via cross-window states.
- **Ablation study (W10):** We have added an ablation on MSM scanning direction.

---

**Reviewer NxX4**

- **Architecture complexity and necessity (W1, Limitations):** Each component is purposefully designed to address sparsity, long sequences, and multi-scale challenges in event data.
- **Limited efficiency gain (W2):** We have clarified the trade-off between representational power and speed, and justified our efficiency metrics.
- **Is MSM augmentation? (Q1):** We position MSM as a multi-view scanning strategy, bridging high-dimensional inputs with 1D Mamba processing.

---

**Reviewer Xb3d**

- **Unclear network description (W1):** We have revised the section to clarify the distinct roles of SEST (serialization) and EventsMamba (feature extraction).
- **Limited ablations (W2):** Ablations on key hyperparameters and MSM have been added.
- **Event subsampling (Q1):** The model samples batches using a fixed number of events without intra-batch downsampling.
- **Theoretical limitations (Limitations):** A new appendix section discusses how the hybrid design addresses structural limitations of GNNs and SSMs.

---

We are committed to implementing all the revisions above in the final version to fully address the concerns raised.

We welcome any further questions or clarifications during this stage and are ready to respond promptly.

Thank you again for your time and thoughtful feedback.

Sincerely,
All authors of submission 27341

---

### Note · Authors · 2025-08-12

**Dear Area Chair,**

We would like to take this opportunity to sincerely thank you for your support throughout the review process, and to also express our gratitude to the reviewers for their valuable time and insightful feedback.

We are greatly encouraged that the in-depth discussions have led to **a positive consensus**. It is particularly gratifying that reviewers have explicitly confirmed that their main concerns have been addressed and have provided **positive final evaluations**. We sincerely value and appreciate this recognition.

In response to these invaluable comments, we have made systematic improvements to the paper, with the main enhancements as follows:
- **Design Justification:** We have supplemented qualitative visualization analyses and discussions on the theoretical limitations of baseline models, offering a deeper explanation of the motivation and value of our architectural design.
- **Methodology:** We have further refined the descriptions of key modules (e.g., micro-level modeling, SEST) and core processes, ensuring full consistency between figures, text, and formulas, thereby significantly improving clarity and reproducibility.
- **Experimental Validation:** We have added sensitivity analyses for key hyperparameters and ablation studies within modules, providing stronger quantitative evidence for the robustness and necessity of our design.

We believe that, through the refinements prompted by this review round, the rigor and completeness of the paper have reached a higher standard. The novelty, efficiency, and representational capability of our proposed EventMG have also been more convincingly substantiated.

We sincerely hope that this conclusion, along with our detailed responses and revisions during the rebuttal phase, will serve as a valuable reference for your final assessment. Once again, we thank you for your time and consideration, and the reviewers for their invaluable guidance. We look forward to your final decision.

Best regards,
*The Authors of Submission 27341*

---

### Decision · Program_Chairs · 2025-09-17

**Decision:**

Accept (poster)

**Comment:**

All reviewers are aligned that the paper be accepted with some swapping to accept after substantial discussion. The AC thanks the reviewers and authors for the detailed discussion and agrees with the reviews that the paper should be accepted to NeurIPs.